# ATTENTIONINFLUENCE: ADOPTING ATTENTION HEAD INFLUENCE FOR WEAK-TO-STRONG PRETRAINING DATA SELECTION

## ABSTRACT

Recently, there has been growing interest in collecting reasoning-intensive pretraining data to improve the reasoning ability of LLMs. Prior approaches typically rely on supervised classifiers to identify such data, requiring labeling by humans or LLMs, often introducing domain-specific biases. Since attention heads are crucial to in-context reasoning, we propose **AttentionInfluence**, a simple yet effective, **training-free** method **without supervision signal**. Our approach enables a **small pretrained language model** to act as a strong data selector through a simple attention head masking operation. Specifically, we identify retrieval heads and compute the loss difference incurred by masking them. We apply AttentionInfluence to a 1.3B-parameter dense model to conduct data selection on the SmolLM corpus of 241B tokens, and mix the corpus with the selected subset comprising 73B tokens to pretrain a 7B-parameter dense model using 1T training tokens and the Warmup-Stable-Decay (WSD) learning rate schedule. Experimental results demonstrate substantial improvements, ranging from **0.8pp** to **3.5pp**, across several knowledge-intensive and reasoning-heavy benchmarks (i.e., MMLU, MMLU-Pro, SuperGPQA, GSM8K, and HumanEval). This demonstrates an effective **Weak-to-Strong** scaling property, with small models improving the performance of larger models—offering a promising and scalable path for reasoning-centric data selection. Code is available.[1]

## 1 INTRODUCTION

The identification of high-quality pretraining data has been a key factor in developing Large Language Models (LLMs). Commonly recognized high-quality pretraining materials include academic papers (e.g., arXiv), books (e.g., Project Gutenberg), high-quality code (e.g., GitHub), and instruction datasets (Li et al., 2024). Existing approaches often rely on manually curated high-quality seed data to train classifiers for extracting additional high-quality pretraining data from massive web corpora. However, as the demand for the scale and diversity of LLMs' pretraining data continues to grow, these carefully curated classifiers suffer from the high manual effort requirements and relatively low diversity of identified data. This raises a critical research question: *How can we continue to identify diverse high-quality pretraining data effectively and scalably?*

Current mainstream methods (Su et al., 2024) typically use supervised or weakly supervised data to train classifiers to identify high-quality data. For instance, Llama 2 (Touvron et al., 2023) uses reference information of Wikipedia documents, which can be seen as weakly supervised data to train a fastText (Joulin et al., 2016) classifier and then recognize Wikipedia-like documents. Llama 3 (Grattafiori et al., 2024) and FineWeb-Edu (Penedo et al., 2024) use LLM-generated responses to train a classifier for educational value, which can be regarded as a much sparser form of distillation from a larger LLM(up to 70B dense parameters) than knowledge distillation (Hinton et al., 2015). While other approaches like DCLM aim to fit user preferences through utilizing signals of user behavior, these methods may introduce potential bias and harm diversity (Li et al., 2024). There are

---

[1] Core implementation of AttentionInfluence is available in an anonymous repository at `https://github.com/gofornlpsota/AttentionInfluence`. The full codebase is under review, but the released core implementation is sufficient to easily and faithfully reproduce all experimental results reported in this paper.

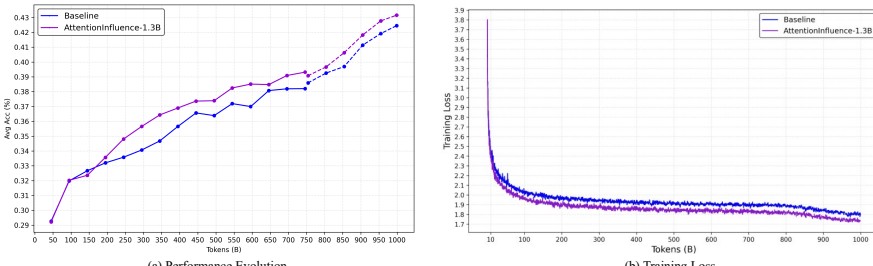

Figure 1: **(a) Performance evolution on comprehensive benchmark evaluations during pretraining.** The first 746 billion tokens correspond to the pretraining phase, represented by solid lines, while the subsequent 254 billion tokens represent the learning rate annealing phase, represented by dashed lines, using the same dataset. After around 200 billion tokens, AttentionInfluence-1.3B consistently outperforms the baseline across a wide range of tasks on average, including the annealing phase. **(b) Training loss during pretraining.** AttentionInfluence-1.3B consistently achieves a lower loss than the baseline.

also efforts to train several domain classifiers and combine them for practical use (Wettig et al., 2025). However, we assume that these methods fail to capture the essence of what makes data reasoning-intensive, and as a result, they can be labor-intensive and require significant data engineering efforts. Moreover, there exists a risk that the classification results from small models distilled from larger models' responses may not improve the final performance of larger models.

Therefore, we propose **AttentionInfluence**, which leverages the intrinsic mechanism of existing LLMs' attention heads for pretraining data selection to achieve weak-to-strong generalization. Existing research suggests that feedforward networks (FFNs) store atomic knowledge (Geva et al., 2020), while attention mechanisms execute algorithms and store procedural knowledge (Olsson et al., 2022; Wu et al., 2024). These mechanistic interpretability insights inspire us to hypothesize that the data activating more important attention heads are high-quality and about procedural knowledge. To be specific, we select the data with a relatively larger loss difference when small pretrained language models process them with and without masking retrieval heads. Compared with mainstream data selection methods (Li et al., 2024; Joulin et al., 2016), AttentionInfluence is training-free and more generalizable.

To validate AttentionInfluence, we adopt a pretrained Llama2-like-1.3B model for data selection from the SmolLM corpus. This 241B-token dataset was already mainly filtered for quality using an education-focused classifier (FineWeb-Edu Classifier). For comparison, we pretrain a 7B dense language model as our baseline on the full SmolLM corpus. Then, we mix the full SmolLM corpus with the high-quality data selected by the 1.3B model to pretrain a 7B dense language model as our model, namely AttentionInfluence-1.3B. As shown in Figure 1, despite the strong baseline, AttentionInfluence-1.3B still yields consistent improvements, demonstrating its ability to further enhance overall data quality through better data selection. Moreover, AttentionInfluence-1.3B shows consistent improvements against the baseline across a wide range of tasks, further demonstrating the effectiveness of the selected data. We further compare AttentionInfluence's selected samples with those of the FineWeb-Edu Classifier. We find that AttentionInfluence selects data that is more balanced, broadly distributed across content categories, and favors longer and more comprehensive samples. Despite being entirely supervision-free and training-free, AttentionInfluence also shows strong agreement with trained-classifier-based patterns, validating its reliability and generalizability.

In summary, our key contributions are as follows:

1. We propose **AttentionInfluence**, a novel **unsupervised** framework that leverages attention head mechanisms to quantify the reasoning intensity for **effective data selection without training any classifiers**.

2. We show that data selected by AttentionInfluence is **high-quality and well-distributed**, thereby yielding consistent improvements across a wide range of tasks.

3. We demonstrate that this approach exhibits **Weak-to-Strong** scaling property, where data selected by a small model significantly improves a larger model's performance.

## 2 RELATED WORK

### 2.1 DATA SELECTION

Many training-free methods use heuristic filtering rules (Rae et al., 2021; Xie et al., 2023b) or perplexity of existing LLMs (Ankner et al., 2024) to assess the quality of pretraining data. For instance, Scaling Filter (Li et al., 2024) evaluates text quality by measuring the perplexity difference between a small and a large language model trained on the same dataset. Some methods leverage weak supervision from Wikipedia-style text to identify high-quality documents (e.g., Llama 2), while others such as DCLM fit user preferences from behavioral signals. In contrast, methods that train models using human-labeled or LLM-generated labels—such as Llama 3, FineWeb-Edu, and ProX—have gained more attention due to their higher accuracy and broader applicability. Recent work (Wettig et al., 2024; Zhao et al., 2024; Peng et al., 2025) further explores multi-class classifiers using data labeled by proprietary commercial LLMs, such as GPT series. There are also efforts to train several domain classifiers (Wettig et al., 2025; M-A-P, 2024) and combine them for practical use. Another line of work focuses on optimizing the data mixture in pretraining corpora, through online and offline frameworks. On the one hand, online approaches (Ye et al., 2024; Xie et al., 2023a) use small proxy models to dynamically reweight data domains during training. On the other hand, offline approaches (Held et al., 2025; OLMo et al., 2024; Liu et al., 2024) train small proxy models on diverse mixtures to identify effective corpus compositions, often using regression or curriculum strategies. AttentionInfluence can be seen as a training-free method without any training cost or data annotation.

### 2.2 MECHANISTIC INTERPRETABILITY

Understanding the inner workings of LLMs is crucial for advancing artificial general intelligence safely. Olsson et al. (2022) and Wu et al. (2024) reveal certain heads are responsible for in-context learning and retrieval, respectively. Lv et al. (2024) further explores how attention heads and MLPs collaborate for factual recall. Sparse autoencoders (Bricken et al., 2023) and head importance estimation (Fu et al., 2024) are also used to analyze or optimize head behaviors. AttentionInfluence adopts a proxy task, proposed by Wu et al. (2024); Qiu et al. (2024), to detect specific important heads, namely the retrieval heads in this paper. AttentionInfluence naturally extends the insights from Wu et al. (2024), broadening their application beyond model analysis and inference acceleration to include effective and efficient data selection.

### 2.3 INFLUENCE MEASURE

Ruis et al. (2024) uses influence functions to recognize pretraining documents important for learning factual knowledge and mathematical reasoning separately. Mirror Influence (Ko et al., 2024) realizes an efficient data influence estimation to select high-quality data. MATES (Yu et al., 2024) continuously adapts a data influence model to the evolving data preferences of the pretraining model and then selects the most effective data for the current pretraining progress. Our work is similar to Mirror Influence in that we use data influence estimation to select high-quality data. However, while Mirror Influence requires a high-quality dataset to train a strong reference model and create a model pair with significant differences in capabilities to compute delta loss, our approach uses the attention mechanism to derive a weaker reference model from the base model. This enables us to obtain two models with a significant capability gap and compute delta loss to evaluate data quality.

## 3 PRELIMINARY

To estimate the impact of each pretraining data sample on LLMs' intrinsic reasoning and retrieval capabilities, we adapt the retrieval score defined in Wu et al. (2024) and model it as a token-level recall rate based on the attention head behavior. We denote the token generated at decoding step $t$ of the LLM as $w_t$. Let the input context have length $n$, and let $t - 1$ tokens have been generated so far. Then the full input sequence at step $t$ is $\mathbf{x}_{1:n+t-1}$. The attention scores of a head at this step are denoted as $\mathbf{a}_t \in \mathbb{R}^{n+t-1}$, i.e., a $1 \times (n + t - 1)$ vector over the full input sequence:

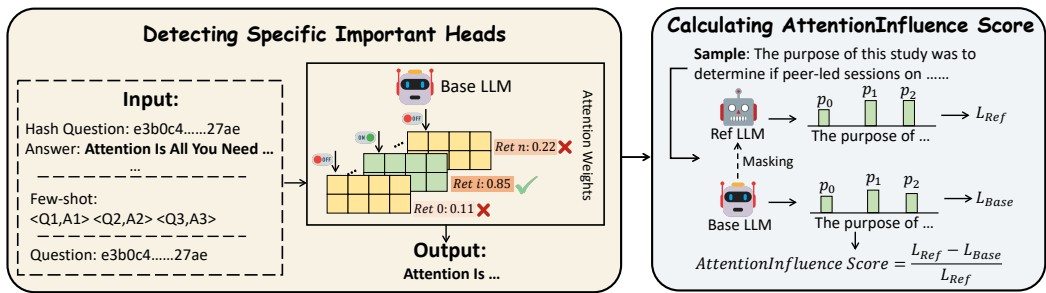

Figure 2: The illustration of AttentionInfluence.

$$\mathbf{a}_t \in \mathbb{R}^{n+t-1}, \quad \mathbf{x}_{1:n+t-1} = [\underbrace{x_1, \ldots, x_n}_{\text{context}}, \underbrace{w_1, \ldots, w_{t-1}}_{\text{generated tokens}}]. \tag{1}$$

We assume that an attention head $h$ performs a copy-paste operation on the corresponding content $\mathbf{k}$ in the context $\mathbf{x}_{1:n}$, i.e.,

$$\mathbf{k} \subseteq \mathbf{x}_{1:n}, \tag{2}$$

if and only if the following two conditions are satisfied:

**Condition 1:** The generated token $w_t$ at decoding step $t$ appears in the corresponding content $\mathbf{k}$:

$$w_t \in \mathbf{k}. \tag{3}$$

**Condition 2:** The token $w_t$ receives the highest attention score among all positions visible to the current query token in this head:

$$j^* = \arg\max_{j \in \{1, \ldots, n+t-1\}} \mathbf{a}_t[j], \quad x_{j^*} \in \mathbf{x}_{1:n+t-1}, \quad x_{j^*} = w_t. \tag{4}$$

Let $\mathbf{g_h}$ denote the set containing all tokens copied and pasted by a given head $h$, we define:

$$\text{Retrieval score for head } h = \frac{|\mathbf{g_h} \cap \mathbf{k}|}{|\mathbf{k}|} \tag{5}$$

## 4 METHOD

Lin et al. (2024) demonstrates that a well-trained reference model can serve as a proxy to fit the desired data distribution of the LLM pretraining by comparing the data loss gap between the base model and the reference model. By comparing the token-level data loss gap between the base model and the reference model, they can identify important tokens that align better with the target distribution. Inspired by recent work lin2024rho, ko2024mirrored, we propose **AttentionInfluence** to select high-quality pretraining data based on the data loss gap from a <weak model, strong model> pair. However, while existing approaches (Lin et al., 2024; Ko et al., 2024) focus on building a stronger reference model as the *strong model*, AttentionInfluence points out that it is cheaper and more controllable to degrade the base model to a weaker version, thus constructing a <weak model, strong model> pair.

Existing studies (Olsson et al., 2022; Wu et al., 2024) point out that specific attention heads (i.e., **retrieval heads**) plays a critical role in LLMs' in-context learning, retrieval, and reasoning capabilities. We find that the language model's retrieval heads emerge early in training, gradually strengthen, and eventually become entrenched in the middle to late stages, playing a crucial role in the model's performance, as shown in Figure 3[2]; further details can be found in Appendix B. Therefore, AttentionInfluence identifies the specific attention heads that are important for targeted LLM capabilities and obtains a degraded reference model by disabling them. Then, AttentionInfluence

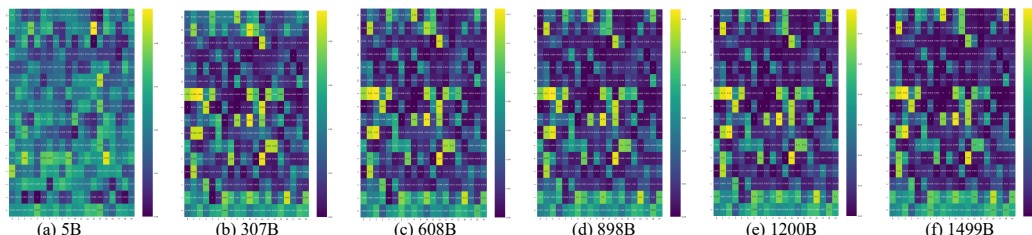

|  |  |  |  |  |  |
|---|---|---|---|---|---|
| (a) 5B | (b) 307B | (c) 608B | (d) 898B | (e) 1200B | (f) 1499B |

Figure 3: The evolution of retrieval heads in a 1.3B dense model.

selects high-quality pretraining data based on the sample-level data loss gap from the constructed <weak model, strong model> pair.

We detail the AttentionInfluence method in the following section.

## 4.1 DETECTING SPECIFIC IMPORTANT HEADS

In this work, we detect the retrieval heads as specifically important heads for reasoning, because Wu et al. (2024) reveals that retrieval heads are extremely relevant to LLMs' retrieval and reasoning ability.

We adopt a Key-Passage Retrieval evaluation task, proposed in CLongEval (Qiu et al., 2024), to evaluate the retrieval ability of LLMs in a controlled setting, and identify attention heads that are strongly associated with retrieval and reasoning. To this end, we construct a synthetic test dataset consisting of 800 samples. Each sample is formatted as a 3-shot retrieval task in natural language, consisting of a `context`, three in-context demonstrations, and a query `hash_key`. The sample template is detailed in Appendix A. Each `context` is a JSON object with $k$ key-value pairs, where each key is a randomly generated 32-character alphanumeric string (`hash_key`), and each value (`text_val`)[3] is a natural language sentence sampled from a corpus of web documents. The task requires the model to retrieve the `text_val` from the `context` and output the `text_val` corresponding to the given query `hash_key`. The inclusion of three in-context demonstrations (i.e., 3-shot) is designed to simulate a few-shot learning scenario and help the model understand the task. Considering the context length limitation of existing pretrained models, we constrain the total length of each test sample—including both the input prompt and the answer—to be close to but not exceeding 4,096 tokens.

Next, we compute retrieval scores for each attention head across test samples, as described in Section 3. In this work, we use a 1.3B-parameter model based on the Llama2-like architecture as the small pretrained language model. We use the average score as the head's final retrieval score and sort them by it. Referring to Wu et al. (2024), we select the heads ranked in the top 5% as specifically important heads. In addition, we conduct ablation studies in Appendix F to examine how different proxy tasks affect the identification of important heads.

## 4.2 CALCULATING ATTENTIONINFLUENCE SCORE

We obtain a reference model by masking the important heads of the base model detected in the first phase, and compute the AttentionInfluence score based on the base model and reference model. For details on the masking operation, refer to Appendix C. First, we use the base model to compute the mean token-level cross-entropy loss ($\mathcal{L}_{\text{base}}$) of each sample in the corpus. Subsequently, we compute the corresponding loss ($\mathcal{L}_{\text{ref}}$) using the reference model. Finally, we use the relative delta between $\mathcal{L}_{\text{base}}$ and $\mathcal{L}_{\text{ref}}$ as an AttentionInfluence Score to quantify the reasoning intensity of each sample, which can be denoted as:

$$\text{AttentionInfluence Score} = \frac{\mathcal{L}_{\text{ref}} - \mathcal{L}_{\text{base}}}{\mathcal{L}_{\text{base}}} \qquad (6)$$

---

[2]The vertical axis corresponds to the transformer layer depth, and the horizontal axis denotes the attention head index within each layer.

[3]Each `text_val` is capped at a maximum of 30 tokens.

Since the loss of a language model for data from different domains (e.g., general/math/code) cannot be directly compared due to significant distribution differences, we restrict the AttentionInfluence Score to be compared only within the same domain (e.g., general/math/code). We consider that a higher AttentionInfluence Score indicates a higher reasoning intensity of the sample.

## 5    EXPERIMENTS AND RESULTS

In this section, we present experimental analyses to validate the effectiveness of the reasoning-intensive data selected by AttentionInfluence.

### 5.1    EXPERIMENTAL DETAILS

We apply AttentionInfluence to a **Llama2-like-1.3B** pretrained model to rank the SmolLM [4] (Ben Allal et al., 2024) corpus. The specifications of the model are described in Appendix G. Specifically, we select the top 20% of samples within each domain in the corpus based on the AttentionInfluence score, yielding approximately **73.1B** reasoning-intensive tokens.

To evaluate the effectiveness of AttentionInfluence, we pretrain a **7B** dense model using a combination of the SmolLM corpus and the selected 73.1B tokens. For comparison, we pretrain another model of identical architecture and size using only the SmolLM corpus, serving as the baseline. Since AttentionInfluence is unsupervised and training-free, we include two unsupervised baselines: (1) a *Perplexity (PPL) Filter*, which selects samples according to their language modeling perplexity (details in Appendix H.1); and (2) a *Scaling Filter* (details in Appendix H.2). To further demonstrate the effectiveness of AttentionInfluence, we include a strong supervised and training-required baseline — the FineWeb-Edu Classifier[5]—distilled from LLaMA2-70B-instruct's responses and serving as an LLM-judge method (details in Appendix H.3).

The model architecture follows that of Llama 2, with detailed hyperparameters listed in Table 7. Detailed information about the SmolLM corpus and pretraining configurations can be found in Appendix G.

Following Grattafiori et al. (2024), we adopt a comprehensive set of benchmark evaluations across **four** major categories in the few-shot setting to holistically compare our model with the baseline: **1) Aggregate Benchmarks**, **2) Math, Code, and Reasoning**, **3) Commonsense Reasoning and Understanding**, and **4) Reading Comprehension**. Detailed descriptions of the benchmarks and evaluation setup are provided in Appendix G.

### 5.2    RESULTS

**Overall Results:** As shown in Figure 1, **AttentionInfluence consistently outperforms the baseline**. Notably, the performance gap emerges early—well before reaching 100B tokens—and becomes both clear and stable throughout training, with AttentionInfluence-1.3B consistently outperforming the baseline on average and across diverse tasks spanning all four benchmark categories. As shown in Table 1, compared to the baseline, AttentionInfluence yields substantial improvements across all four benchmark categories, with gains ranging from **0.8pp** to **3.5pp** on various tasks. Furthermore, during the middle stage of training, our unsupervised and training-free AttentionInfluence matches the performance of the strong supervised and training-required FineWeb-Edu Classifier and surpasses all other unsupervised baselines[6], ultimately outperforming all methods upon completion of the full 1T-token training as shown in Table 2. The full results, encompassing all methods evaluated throughout the training stages, are provided in Table 9.

---

[4]https://github.com/huggingface/smollm/tree/main/text/pretraining

[5]https://huggingface.co/HuggingFaceFW/fineweb-edu-classifier

[6]Due to limited computational resources, the training of other unsupervised baselines was halted at the middle of pretraining, as they had already fallen behind the strong supervised FineWeb-Edu Classifier.

| Model | #Tokens | Avg. | Metrics | | | | | | | |
|---|---|---|---|---|---|---|---|---|---|---|
| Baseline w/o LRD | 495B | 36.39 | ARC-C 54.35 | ARC-E 81.44 | ARC(C+E) 67.89 | Wino. 64.40 | Hella. 71.21 | CSQA 32.19 | OpenBookQA 46.20 | PIQA 78.02 |
| | | | TriviaQA 43.74 | MMLU 35.44 | MMLU-Pro 13.12 | AGIEval-en 20.59 | GPQA 22.23 | SuperGPQA 9.44 | RACE 39.52 | DROP 28.93 |
| | | | BBH 32.29 | GSM8K 12.05 | MATH 6.08 | HumanEval 19.94 | C-Eval 25.48 | CMMLU 27.42 | | |
| PPL filter w/o LRD | 495B | 36.54 | ARC-C 53.07 | ARC-E 80.35 | ARC(C+E) 66.71 | Wino. 65.51 | Hella. 70.73 | CSQA 39.97 | OpenBookQA 44.40 | PIQA 78.40 |
| | | | TriviaQA 43.69 | MMLU 39.53 | MMLU-Pro 13.33 | AGIEval-en 20.80 | GPQA 22.30 | SuperGPQA 9.20 | RACE 39.43 | DROP 27.71 |
| | | | BBH 29.50 | GSM8K 9.10 | MATH 5.16 | HumanEval 20.27 | C-Eval 28.10 | CMMLU 26.70 | | |
| Scaling Filter w/o LRD | 495B | 36.81 | ARC-C 52.65 | ARC-E 81.31 | ARC(C+E) 66.98 | Wino. 63.54 | Hella. 70.43 | CSQA 40.62 | OpenBookQA 42.80 | PIQA 77.48 |
| | | | TriviaQA 44.05 | MMLU 39.37 | MMLU-Pro 13.81 | AGIEval-en 21.20 | GPQA 24.90 | SuperGPQA 9.63 | RACE 39.71 | DROP 28.69 |
| | | | BBH 30.60 | GSM8K 11.60 | MATH 5.80 | HumanEval 18.75 | C-Eval 28.50 | CMMLU 27.60 | | |
| FineWeb-Edu Classifier w/o LRD | 495B | 37.44 | ARC-C 54.35 | ARC-E 81.73 | ARC(C+E) 68.04 | Wino. 64.96 | Hella. 70.34 | CSQA 46.60 | OpenBookQA 44.00 | PIQA 77.58 |
| | | | TriviaQA 43.17 | MMLU 41.00 | MMLU-Pro 13.36 | AGIEval-en 20.46 | GPQA 22.94 | SuperGPQA 9.36 | RACE 40.67 | DROP 30.08 |
| | | | BBH 30.94 | GSM8K 12.51 | MATH 7.10 | HumanEval 18.66 | C-Eval 28.45 | CMMLU 27.97 | | |
| AttentionInfluence-1.3B w/o LRD | 495B | 37.39 | ARC-C 52.13 | ARC-E 80.35 | ARC(C+E) 66.24 | Wino. 65.19 | Hella. 71.40 | CSQA 44.39 | OpenBookQA 45.20 | PIQA 77.09 |
| | | | TriviaQA 43.43 | MMLU 39.72 | MMLU-Pro 14.38 | AGIEval-en 21.51 | GPQA 24.26 | SuperGPQA 10.04 | RACE 39.04 | DROP 29.88 |
| | | | BBH 33.45 | GSM8K 12.51 | MATH 6.05 | HumanEval 17.87 | C-Eval 27.93 | CMMLU 29.37 | | |
| AttentionInfluence-7B w/o LRD | 495B | 37.96 | ARC-C 51.28 | ARC-E 79.55 | ARC(C+E) 65.42 | Wino. 65.04 | Hella. 71.29 | CSQA 52.42 | OpenBookQA 44.60 | PIQA 78.18 |
| | | | TriviaQA 44.49 | MMLU 42.64 | MMLU-Pro 15.66 | AGIEval-en 22.74 | GPQA 21.22 | SuperGPQA 10.73 | RACE 38.28 | DROP 29.94 |
| | | | BBH 32.25 | GSM8K 13.42 | MATH 6.05 | HumanEval 18.63 | C-Eval 29.72 | CMMLU 29.12 | | |

Table 1: Main results on various benchmarks at the middle stage of training ( 500B tokens). The LRD denotes learning rate decay.

| Model | #Tokens | Avg. | Metrics | | | | | | | |
|---|---|---|---|---|---|---|---|---|---|---|
| Baseline w/ LRD | 1T | 42.46 | ARC-C 58.79 | ARC-E 83.92 | ARC(C+E) 71.36 | Wino. 70.24 | Hella. 75.63 | CSQA 59.62 | OpenBookQA 48.00 | PIQA 80.63 |
| | | | TriviaQA 51.07 | MMLU 50.05 | MMLU-Pro 19.32 | AGIEval-en 27.06 | GPQA 24.77 | SuperGPQA 12.10 | RACE 41.15 | DROP 36.09 |
| | | | BBH 35.42 | GSM8K 21.00 | MATH 8.74 | HumanEval 23.02 | C-Eval 33.80 | CMMLU 31.33 | | |
| FineWeb-Edu Classifier w/ LRD | 1T | 42.66 | ARC-C 57.85 | ARC-E 83.67 | ARC(C+E) 70.76 | Wino. 68.03 | Hella. 75.21 | CSQA 61.59 | OpenBookQA 47.00 | PIQA 80.09 |
| | | | TriviaQA 49.93 | MMLU 51.92 | MMLU-Pro 20.76 | AGIEval-en 30.27 | GPQA 25.99 | SuperGPQA 12.12 | RACE 41.82 | DROP 34.68 |
| | | | BBH 35.97 | GSM8K 20.62 | MATH 10.00 | HumanEval 24.36 | C-Eval 32.54 | CMMLU 31.45 | | |
| AttentionInfluence-1.3B w/ LRD | 1T | 43.16 | ARC-C 59.98 | ARC-E 84.26 | ARC(C+E) 72.12 | Wino. 68.03 | Hella. 75.49 | CSQA 61.59 | OpenBookQA 46.60 | PIQA 79.54 |
| | | | TriviaQA 51.20 | MMLU 51.48 | MMLU-Pro 22.03 | AGIEval-en 27.30 | GPQA 24.26 | SuperGPQA 12.92 | RACE 42.30 | DROP 36.52 |
| | | | BBH 36.80 | GSM8K 23.73 | MATH 10.00 | HumanEval 26.55 | C-Eval 33.06 | CMMLU 32.75 | | |

Table 2: Main results on various benchmarks after full training (1T tokens). The LRD denotes learning rate decay.

**AttentionInfluence Remarkably Enhances LLMs' Comprehensive Knowledge:** On challenging aggregate benchmarks such as MMLU, MMLU-Pro, and AGIEval-en, AttentionInfluence consistently outperforms the baseline, indicating stronger comprehensive knowledge and reasoning capabilities. Improvements of **+1.4pp** on MMLU, **+2.7pp** on MMLU-Pro, and **+0.8pp** on SuperGPQA clearly demonstrate the effectiveness of AttentionInfluence in selecting diverse pretraining data that supports both **broad knowledge acquisition** and **reasoning-intensive learning**.

**AttentionInfluence Brings Significant Improvements for Complex Reasoning Tasks:** Attention-Influence yields substantial improvements on complex multi-step reasoning tasks such as GSM8K (**+2.7pp**), MATH (**+1.3pp**), HumanEval (**+3.5pp**), and BBH (**+1.4pp**), suggesting that the selected data distribution better facilitates problem-solving and advanced reasoning. Additional gains on ARC-Challenge, DROP, and RACE further demonstrate that **AttentionInfluence enhances reasoning generalization across a wide range of tasks**.

## 6 DISCUSSION

### 6.1 RELIABILITY OF ATTENTIONINFLUENCE

To validate the effectiveness of AttentionInfluence, we design two metrics—**Education Score** and **Reasoning Score**—to quantify the quality of the selected data. Specifically, we randomly sample 200 examples from the top 20% ranked by AttentionInfluence and the FineWeb-Edu classifier, respectively, and employ GPT-4o (Achiam et al., 2023; Hurst et al., 2024) as the evaluator. Detailed scoring criteria and prompt design for both metrics are provided in Appendix J.

As shown in Table 10, both AttentionInfluence and the FineWeb-Edu classifier yield comparable scores on education-related content. However, AttentionInfluence achieves substantially higher scores in reasoning, indicating that **samples selected by AttentionInfluence exhibit greater reasoning intensity**.

Additionally, we analyze the length of the selected samples and find that AttentionInfluence consistently selects longer samples on average across domains than the FineWeb-Edu classifier. In the Python-Edu and OpenWebMath domains, AttentionInfluence selects samples with an average length nearly twice that of those selected by the FineWeb-Edu classifier. A qualitative inspection of these samples (see Appendix L) reveals that, in the Python-Edu domain, AttentionInfluence favors documents that contain not only more complex code but also richer textual context, such as in-depth programming tutorials that offer detailed explanations of the code. In the OpenWebMath domain, samples selected by AttentionInfluence demonstrate more elaborate formula-based reasoning. **These findings suggest that AttentionInfluence effectively identifies data with more comprehensive and complex reasoning structures.**

### 6.2 DIVERSITY OF SELECTED DATA BY ATTENTIONINFLUENCE

#### 6.2.1 CLUSTERING-BASED DISTRIBUTION ANALYSIS

To better understand the distribution of samples selected by different methods (i.e., AttentionInfluence and the FineWeb-Edu classifier), we perform clustering on the selected subsets and employ GPT-4o to annotate the resulting clusters. The clustering procedure is detailed in Appendix K.

We derive the following insights:

**1) AttentionInfluence produces a more balanced distribution across data categories.** As illustrated in Figure 11, both methods cover a broad range of top-level categories. However, the distribution from AttentionInfluence is notably more balanced.

**2) AttentionInfluence selects a highly diverse set of samples.** We examine two clusters from the AttentionInfluence subset that exhibit large embedding distances. As illustrated by examples from the *Health Guidelines & Nutrition* and *Information Technology* clusters in Appendix M, the selected samples differ substantially in both content and style. This semantic divergence underscores the effectiveness of the clustering and further enhances the interpretability of the annotated categories.

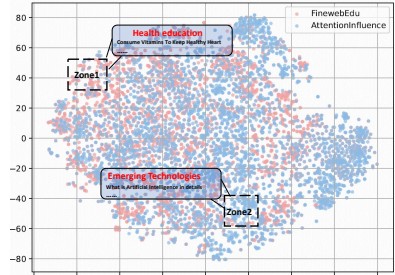

Figure 4: Data selected by Attention-Influence and FineWeb-Edu Classifier.

#### 6.2.2 THE VISUALIZATION OF DATA DISTRIBUTION

To intuitively illustrate the distributions of samples selected by the two methods, we apply Principal Component Analysis (PCA) to reduce the dimensionality of document embeddings and visualize the results in two-dimensional space.

As shown in Figure 4, AttentionInfluence selects samples with broader and more balanced coverage. **By directly leveraging the attention mechanisms of pretrained language models, it facilitates more effective selection of general and diverse training data than the FineWeb-Edu classifier.**

**In addition, the selected samples from the two methods exhibit complementary coverage.** We further examine the distinctive regions covered by AttentionInfluence and the FineWeb-Edu classifier. For example, the samples in Zone1 are related to Health Education, while most samples in Zone2 fall under the theme of Emerging Technologies. This suggests that samples selected by the two methods can be complementary. How to effectively integrate the strengths of both data selection strategies could be a promising direction for future exploration.

### 6.3 SCALABILITY OF ATTENTIONINFLUENCE

We compare the samples selected by the AttentionInfluence method using 1.3B and 7B pretrained language models. We obtain the following insights:

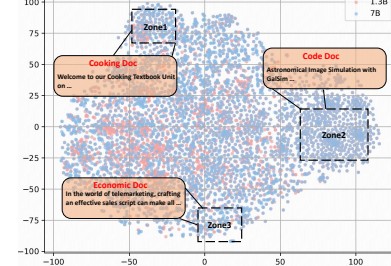

**1) AttentionInfluence based on a larger LLM selects higher-quality data.** Similar to the setting in Section 6.1, we use GPT-4o to evaluate selected samples. As shown in Table 11, across all domains, samples selected by the 7B model exhibit higher education scores than those selected by the 1.3B model. Regarding reasoning scores, the 7B model significantly outperforms the 1.3B model across all four domains, with a particularly notable improvement of 9 percentage points in the FineWeb-Edu-dedup domain. These results suggest that larger models are more effective at identifying reasoning-intensive samples. Moreover, we also trained a 7B model on the data selected by AttentionInfluence-7B. As shown in Table 8, the LLM trained on data selected by AttentionInfluence-7B perform better than that trained on data selected by AttentionInfluence-1.3B, which further demonstrates the scalability of AttentionInfluence.

Figure 5: Data selected by Attention-Influence-1.3B/7B.

**2) AttentionInfluence based on a larger LLM is more generalizable.** As shown in the Figure 5, we compare the distributions of samples selected by the 1.3B and 7B models. We observe that samples selected by the 7B model are more broadly distributed, covering many regions that the 1.3B model fails to reach. Notably, regions underrepresented by the 1.3B model are densely populated with specific categories of samples, which are predominantly captured by the 7B model.

For instance, Zone1 corresponds to cooking, Zone2 relates to code, and Zone3 primarily focuses on the economy. This suggests that, even without additional training, samples selected by larger models are more balanced and diverse, capturing a broader range of information. As shown in the appendix (see Table 8, Figure 7, Figure 9 and Figure 10), AttentionInfluence-7B consistently outperforms AttentionInfluence-1.3B across various benchmarks during the middle and later stages of training.

Nonetheless, the performance narrows during the final learning rate annealing phase likely due to saturation in the SmolLM corpus and training setup, as suggested by comparisons with SmolLM (Ben Allal et al., 2024) and SmolLM2 (Allal et al., 2025). Importantly, the selected evaluation benchmarks may not fully capture the generalization benefits of AttentionInfluence-7B. For example, while the SmolLM corpus is predominantly English with minimal Chinese content, we observe that AttentionInfluence-7B significantly outperforms AttentionInfluence-1.3B on the Chinese C-Eval benchmark (see Figure 10), reflecting a broader and more robust generalization capability that remains underexplored under the current evaluation settings.

## 7 CONCLUSION

In this paper, we propose AttentionInfluence, an unsupervised and training-free framework for selecting high-quality and reasoning-intensive pretraining data by leveraging attention head mechanisms in pretrained language models. Experimental results on the SmolLM corpus demonstrate that AttentionInfluence consistently improves LLMs' performance on various benchmarks, selects longer and more diverse data of high quality, and aligns well with the trained-classifier-based selection pattern—while offering promising Weak-to-Strong generalization. Our findings suggest that internal model mechanisms can serve as reliable indicators of data quality, offering a scalable and effective path for LLM pretraining data selection.

ETHICS STATEMENT

We have adhered to the ICLR Code of Ethics in conducting this research. Our work introduces AttentionInfluence, a novel unsupervised method for data selection aimed at enhancing the reasoning capabilities of large language models. We outline the primary ethical considerations associated with our methodology and its potential applications below.

1. Bias in Data Selection Our method, AttentionInfluence, utilizes a small pretrained language model as an unsupervised data selector. A significant consideration is that any societal biases (e.g., regarding gender, race, or culture) inherent in this small selector model could be amplified in the selected data subset. Pretraining a larger model on this subset may consequently entrench or even exacerbate these biases. We acknowledge this limitation and suggest that future work could explore integrating bias mitigation techniques directly into the data selection process to foster greater fairness in the resulting models.

2. Dual Use of Enhanced Reasoning Models As with any research that advances the capabilities of AI, improving the reasoning abilities of LLMs carries a risk of dual use. While our intention is to advance scientific understanding and create more helpful AI systems, we recognize that more powerful reasoning models could potentially be misappropriated for malicious purposes, such as generating sophisticated disinformation or automating harmful tasks. We support the ongoing community-wide dialogue on the responsible development, governance, and deployment of AI technologies to mitigate such risks.

3. Data Privacy Our experiments utilize the SmolLM corpus, a large-scale open-source dataset. Like many corpora scraped from the public web, it may contain personally identifiable information (PII). Our unsupervised data selection method does not inherently identify or remove such sensitive information. The use of publicly available corpora that may contain PII is a broader challenge in the field that warrants continued attention and the development of better data anonymization and curation practices.

4. Responsible Release of Research Artifacts Our primary release consists of the source code for our AttentionInfluence method, made available under a permissive open-source license. We are not releasing the selected data subset concurrently with this publication. However, to further facilitate research, we are open to considering a future release of this subset if there is significant community interest. Any such release would be preceded by a rigorous screening process to mitigate risks related to privacy, bias, and harmful content, in accordance with best practices for responsible dataset publication.

REPRODUCIBILITY STATEMENT

We are committed to ensuring the reproducibility of our research. To facilitate this, we have released the core implementation of our method, AttentionInfluence, along with the code for all baselines presented in this paper. This is provided both as an anonymous and public GitHub repository and as a compressed archive in the supplementary materials. While the full codebase is currently undergoing an internal review, we are confident that the released core implementation is sufficient to easily and faithfully reproduce all experimental results reported in this paper.

Comprehensive details to support reproducibility are provided in the appendices. Specifically, Appendix G details our complete experimental setup, including training data, model architecture, training parameters, and evaluation procedures. Appendix H provides the core implementation details for all the baselines. For our qualitative and quantitative analyses, implementation specifics for the LLM-as-a-judge experiments are located in Appendix J, and details of the clustering analysis are in Appendix K. We plan to release the complete codebase, including all analysis scripts, upon the completion of the internal review process. Furthermore, to fully support community research, we also plan to release the high-quality data subsets separately selected by AttentionInfluence and each baseline method, as well as our intermediate and final trained model checkpoints.

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

## A  SYNTHETIC TEST SAMPLE

```
model input:
Please extract the value corresponding to the specified key from
the following JSON object. Output only the value of the
corresponding key and nothing else. The JSON data is as follows:
{context}

{question-shot1}
{answer-shot1}

{question-shot2}
{answer-shot2}

{question-shot3}
{answer-shot3}

{question}
answer: {answer}
```

## B  EVOLUTION OF RETRIEVAL HEADS IN PRETRAINED MODELS

We apply the method described in Section 4.1 to identify retrieval heads at six checkpoints of the pretrained 1.3B-parameter model. These checkpoints correspond to training progress at 5B, 307B, 608B, 898B, 1200B, and 1499B tokens, respectively. We also analyze the 7B-parameter model, using checkpoints corresponding to training progress at 9B, 1800B, 3600B, 5628B, 7204B, and 8964B tokens, as shown in Figure 6. We observe similar trends to those in the 1.3B model, with retrieval heads exhibiting early emergence and becoming ever more entrenched as training advances. In Figure 3 and Figure 6, the vertical axis corresponds to the transformer layer depth, and the horizontal axis denotes attention head index within each layer.

## C  MASKING OPERATION

The "mask" operation is to set the attention weights provided by the specific attention heads to equal weights. And if the length of the sequence is $L$, the attention weight of each token should be set to $\frac{1}{L}$.

## D  EFFECT OF MASKING RETRIEVAL HEADS VS. RANDOM NON-RETRIEVAL HEADS

The 3-shot Retrieval task corresponds to the proxy task introduced in Section 4.1. Banking77-ICL is an internal evaluation task for assessing a model's in-context learning ability. It requires models to perform many-shot classification on the Banking77 dataset (Casanueva et al., 2020). Here, "Masked, Retrieval Heads" refers to masking attention heads ranked in the top 5% by retrieval score, while

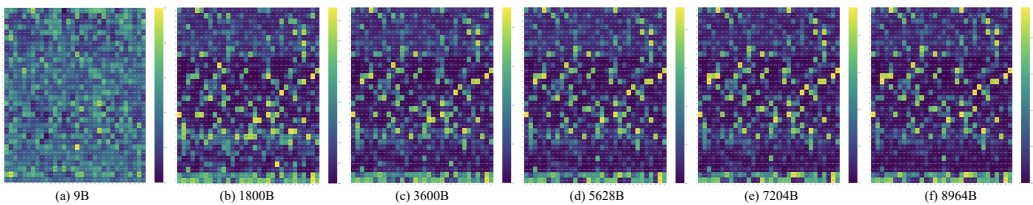

(a) 9B      (b) 1800B      (c) 3600B      (d) 5628B      (e) 7204B      (f) 8964B

Figure 6: The evolution of retrieval heads in a 7B dense model.

"Random Masked, Non-Retrieval Heads" refers to randomly masking heads ranked between the top 5% and top 100% (i.e., the remaining 95%) by retrieval score. We conduct the experiments using the models shown in the Table 5 and find that masking retrieval heads significantly impairs the model's reasoning performance, while masking random non-retrieval heads has only a minor effect—consistent with the findings of Wu et al. (2024). In addition, we find that retrieval heads also play an essential role in the model's in-context learning ability, which may suggest a high overlap between retrieval heads and induction heads.

## E  MIRROR EFFECTS IN ATTENTIONINFLUENCE

For tasks with performance gains—such as MMLU, MMLU-Pro, AGIEval-en, DROP, BBH and GSM8K—we observe that masking retrieval heads in the pretrained 1.3B model leads to a significant performance drop (see Table 3). This suggests a mirror effect: when the performance of the 1.3B model significantly degrades on certain tasks due to masking certain important heads, the data selected by AttentionInfluence-1.3B tends to improve performance on these same tasks when used to train a 7B model. This observation supports the insight discussed in Section 4, demonstrating the interpretability of AttentionInfluence and its predictive power in identifying evaluation metrics likely to show improvements prior to any training.

| Model | Benchmarks | | | | | |
|---|---|---|---|---|---|---|
| 1.3B | Hellaswag 0.5715 | WinoGrande 0.6062 | MMLU 0.4258 | MMLU-Pro 0.1290 | AGIEval-en 0.2047 | GPQA 0.2203 |
| | DROP 0.2344 | BBH 0.3166 | GSM8K 0.1820 | HumanEval 0.1707 | 3-shot Retrieval 0.4213 | Banking77-ICL 0.4148 |
| 1.3B (Random Masked, Non-Retrieval Heads) | Hellaswag 0.5518 | WinoGrande 0.6069 | MMLU 0.4165 | MMLU-Pro 0.1275 | AGIEval-en 0.2072 | GPQA 0.2071 |
| | DROP 0.2190 | BBH 0.3005 | GSM8K 0.1274 | HumanEval 0.1159 | 3-shot Retrieval 0.3838 | Banking77-ICL 0.3840 |
| 1.3B (Masked, Retrieval Heads) | Hellaswag 0.5493 | WinoGrande 0.5801 | MMLU 0.3089 | MMLU-Pro 0.0305 | AGIEval-en 0.1298 | GPQA 0.1827 |
| | DROP 0.1141 | BBH 0.0429 | GSM8K 0.0068 | HumanEval 0.1098 | 3-shot Retrieval 0 | Banking77-ICL 0.0001 |

Table 3: Effect of Masking Retrieval Heads vs. Random Non-Retrieval Heads on Reasoning and In-Context Learning

## F  ABLATION STUDIES ON THE IDENTIFICATION OF IMPORTANT HEADS

The choice of the task for reasoning-head detection is important. We use a JSON key-value extraction task due to its highly controllable structure and its nature as an in-context retrieval task decoupled from prior knowledge, which effectively activates retrieval heads without interference from the training data (e.g., the model having memorized relevant content or specific samples). In future work, we intend to investigate other tasks such as multi-hop question answering or mathematical reasoning to assess the robustness and generality of the identified heads and improve sample selection quality.

To better understand the influence of the proxy task, we conduct an ablation study comparing which heads are selected as retrieval heads under different tasks. Specifically, we reproduced the needle task and implementation used in Wu et al. (2024), referred to here as Plain Needle, and the Reasoning Needle task and implementation from Fu et al. (2024). Using the 1.3B dense checkpoint mentioned in our paper, we applied our method (JSON key-value extraction), plain needle, and reasoning needle to identify the top 5% of heads as retrieval heads, then measured the overlaps. The overlap ratios are summarized in the table below:

| Methods Compared | Overlap Ratio (%) |
|---|---|
| Our Method & Plain Needle | 70.6 |
| Our Method & Reasoning Needle | 29.4 |
| Plain Needle & Reasoning Needle | 11.8 |

Table 4: Overlap ratios of retrieval heads identified under different proxy tasks.

These results suggest that different proxy tasks highlight different types of heads that play key roles within the specific settings of each proxy task. Accordingly, we hypothesize that these heads, when used for data selection, also capture different types of training samples. In addition, we conducted internal experiments to compare pretraining outcomes using data selected by our method versus data selected using the selected heads by Reasoning Needle. The results showed that the latter led to greater improvements on reasoning benchmarks, as expected, though it underperformed slightly in other dimensions compared to our method, yielding overall comparable performance. Due to policy restrictions, we are unable to disclose the exact evaluation metrics from these internal experiments. We plan to further extend this analysis by including reasoning needle–based experiments and results on the SmolLM corpus in future work.

## G EXPERIMENT SETTING

**Pretraining Data**  To ensure reproducibility, we use the SmolLM corpus (Ben Allal et al., 2024) as the pretraining dataset. The composition of the SmolLM Corpus dataset is shown in the Table 6. We sample 100 million tokens from SmolLM corpus as the validation dataset.

**Pretrained models used by AttentionInfluence**  In this work, AttentionInfluence employs internal pretrained models based on the Llama2-like architecture. The hyperparameters of the models are detailed in Table 5.

| model size | pretraining tokens | vocab size | hidden size | ffn inner | num heads | num layers | shared q_head | seq len | tie emb |
|---|---|---|---|---|---|---|---|---|---|
| 1.3B | 1.5TB | 155136 | 2,560 | 10,240 | 20 | 16 | 2 | 4,096 | true |
| 7B | 9TB | 155136 | 4,096 | 16,384 | 32 | 32 | 2 | 8,192 | true |

Table 5: Hyperparams of the Pretrained Models Used by AttentionInfluence.

**Computation Cost of AttentionInfluence on SmolLM corpus**  Using the 1.3B model, we compute AttentionInfluence scores for all samples in the SmolLM corpus (241B tokens) using 128 A100 GPUs (16 machines, each with 8 A100-80GB GPUs and 900GB of CPU memory), which takes approximately 5 hours. For the 7B model, the same computation requires 160 A100 GPUs (20 machines, each with 8 A100-80GB GPUs and 900GB of CPU memory) and takes around 25 hours.

**Model trained in the experiment**  The hyperparameters are presented in Table 7, and tokenizer used for training and computing token counts is the same as SmolLM[7] with a vocab size of 49,152.

**Pretraining setting**  Following SmolLM (Ben Allal et al., 2024), our experiments adopt the WSD learning rate scheduler (Hu et al., 2024), with 0.1% warmup steps, 75% steady phase, and a final 25% decay phase. We use the AdamW optimizer (Loshchilov and Hutter, 2017). Pretraining is conducted on 32 machines, each equipped with 8 H100-80GB GPUs and 2800GB of CPU memory. Each experiment runs for 96 hours, using a total of 1 TB of training tokens—comprising 750B tokens during the constant learning rate phase and 250B tokens during the learning rate decay (annealing) phase.

| Dataset | FineWeb-Edu-dedup | Cosmopedia-V2 | Python-Edu | OpenWebMath |
|---|---|---|---|---|
| # Tokens (billions) | 193.3 | 27.9 | 3.8 | 13.3 |

Table 6: Composition of the SmolLM Corpus Dataset.

**Benchmarks**  We evelute the performance of LLMs across 4 domains: **1) Aggregate Benchmarks**, including AGIEval-en (Zhong et al., 2023), MMLU (Hendrycks et al., 2020), MMLU-Pro (Wang et al., 2024), GPQA (Rein et al., 2023), SuperGPQA (Du et al., 2025), C-Eval (Huang et al., 2023) and CMMLU (Li et al., 2023); **2) MATH, Code, and Reasoning**, comprising GSM8K (Cobbe et al., 2021), MATH (Hendrycks et al., 2021), HumanEval (Chen et al., 2021), ARC Challenge (Clark

---

[7]https://huggingface.co/HuggingFaceTB/cosmo2-tokenizer

| model size | batch size | learning rate | hidden size | ffn inner | num heads | num layers | shared q_head | seq len | tie emb | total params |
|---|---|---|---|---|---|---|---|---|---|---|
| 7B | 1,024 | 4e-4 | 4,096 | 8,192 | 32 | 32 | 4 | 8,192 | false | 6.98B |

Table 7: Hyperparams of the Model Trained in the Experiment.

et al., 2018), DROP (Dua et al., 2019), and BBH (Suzgun et al., 2022); **3) Commonsense Reasoning and Understanding**, including HellaSwag (Zellers et al., 2019), ARC-Easy (Clark et al., 2018), WinoGrande (Sakaguchi et al., 2021), CommonSenseQA (Talmor et al., 2018), PiQA (Bisk et al., 2020), OpenBookQA (Mihaylov et al., 2018), and TriviaQA (Joshi et al., 2017); and **4) Reading Comprehension**, represented by RACE (Lai et al., 2017).

**Evaluation details**   To ensure that all demonstrations, along with the question and the generated prediction, fit within the 8192-token context window, we use a different number of few-shot examples per evaluation task. Specifically, we use the following numbers of demonstrations (in parentheses): MATH (`minerva_math`) (4), DROP (3), BBH (3), GPQA (3), SuperGPQA (3), and 5 for all other tasks. We report accuracy for most tasks, with the following exceptions: `exact_match` for MMLU-Pro, TriviaQA, and BBH; `flexible-extract` for GSM8K; and F1 score for DROP. When available, we use the normalized accuracy (`acc_norm`) metric provided by the lm-evaluation-harness. ARC(C+E) denotes the average accuracy over ARC-Challenge (ARC-C) and ARC-Easy (ARC-E). For specific tasks, we adopt the following exceptions:

- For AGIEval, we conduct the official few-shot-CoT evaluation using the official repository[8].

- For C-Eval and CMMLU, we conduct the official 5-shot evaluation using the official repository [9][10], respectively.

- For GPQA and SuperGPQA, we use an internal evaluation framework, with the common 3-shot-CoT setting.

- For DROP, we fix a known bug in the lm-evaluation-harness implementation, following the discussion[11].

- For BBH, we find that the answer parsing in the lm-evaluation-harness is not entirely accurate, which makes a slight difference. Therefore, we use an internal evaluation framework to assess BBH, with the common 3-shot-CoT setting.

- For MATH, we find that the answer parsing in the lm-evaluation-harness is not entirely accurate. Therefore, we use an internal evaluation framework to assess MATH, with the common 4-shot-CoT setting.

- For HumanEval, we conduct zero-shot evaluation using the BigCode evaluation harness[12] and report pass@1 using the following generation settings, which are the same as those used in SmolLM (Ben Allal et al., 2024): temperature = 0.2, top-p = 0.95, `n_samples` = 20, and `max_length_generation` = 1024.

# H   BASELINE IMPLEMENTATION DETAILS

This appendix provides implementation details for the two unsupervised baselines—the PPL Filter, and Scaling Filter—as well as the supervised baseline, FineWeb-Edu Classifier, used in our experiments. For the Scaling Filter and FineWeb-Edu Classifier, we rank the corpus using the scores produced by each model, following the procedure in Section 5.1, and select the top samples totaling 73.1B tokens. For the PPL Filter, we instead sample from medium-perplexity examples to reach the same total of 73.1B tokens.

---

[8]https://github.com/ruixiangcui/AGIEval/tree/main

[9]https://github.com/SJTU-LIT/ceval

[10]https://github.com/haonan-li/CMMLU

[11]https://github.com/EleutherAI/lm-evaluation-harness/issues/2137

[12]https://github.com/bigcode-project/bigcode-evaluation-harness

### H.1    PERPLEXITY (PPL) FILTER

The *Perplexity (PPL) Filter* selects samples based on their language modeling perplexity, computed with Qwen3-1.7B-Base[13](Team, 2025). Samples are first ranked by perplexity, and those within the 20%-80% range are then sampled to reach a total of 73.1B tokens. We hypothesize that mid-perplexity samples offer higher learning efficiency.

### H.2    SCALING FILTER

We use Qwen3-0.6B-Base[14] as the small model and Qwen3-1.7B-Base as the large model(Team, 2025), and implement the scorer following the method described in Li et al. (2024).

### H.3    FINEWEB-EDU CLASSIFIER

We use the score output by the FineWeb-Edu Classifier to rank the corpus using the same procedure as in  Section 5.1, and select the top samples that also sum up to 73.1B tokens.

## I    DETAILED PERFORMANCE EVOLUTION DURING PRETRAINING

As shown in Figure 7, Figure 9, and Figure 10, we illustrate how the performance of the baseline, the 1.3B method, the 7B method and the FineWeb-Edu Classifier method evolves across different benchmarks as the number of training tokens increases.

In addition, panel (b) of Figure 1 and Figure 8 present the training loss comparison among them. Furthermore, we report the detailed evaluation results of LLMs trained on data selected by AttentionInfluence-1.3B and AttentionInfluence-7B, as shown in Table 8.

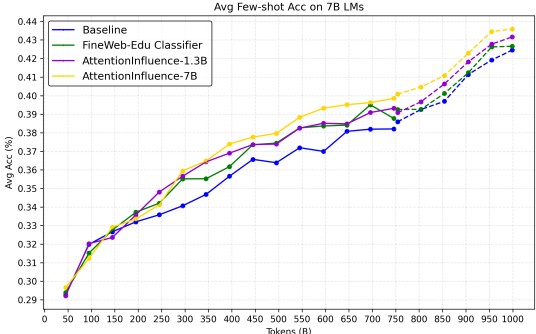

Figure 7: Performance evolution on comprehensive benchmark evaluations during pretraining. The first 746B tokens correspond to the pretraining phase, represented by solid lines, while the subsequent 254B tokens represent the learning rate annealing phase, represented by dashed lines, using the same dataset. Once training surpasses 350B tokens, AttentionInfluence-7B exhibits consistently superior average performance over AttentionInfluence-1.3B, the baseline, and the FineWeb-Edu Classifier across a broad range of tasks, even during the annealing phase.

---

[13]https://huggingface.co/Qwen/Qwen3-1.7B-Base
[14]https://huggingface.co/Qwen/Qwen3-0.6B-Base

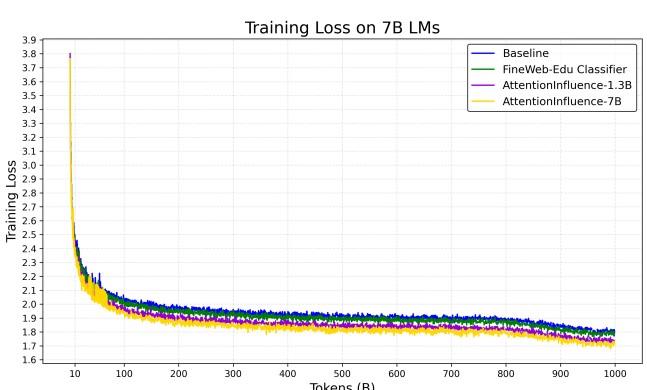

Figure 8: Training loss

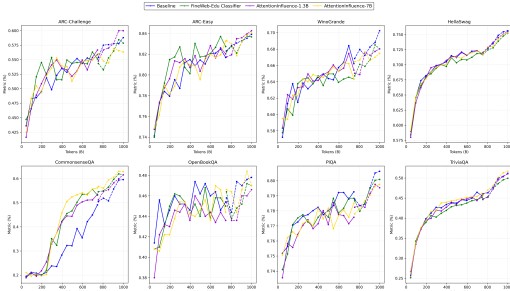

Figure 9: The performance evolution during pretraining on relatively simple benchmarks (i.e., ARC-Challenge, ARC-Easy, WinoGrande, HellaSwag, CommonsenseQA, OpenBookQA, PIQA, TirvialQA). The first 746B tokens correspond to the standard pretraining phase (solid lines), followed by 254B tokens under learning rate annealing (dashed lines). Curves with the same color (solid and dashed) indicate training on the same dataset.

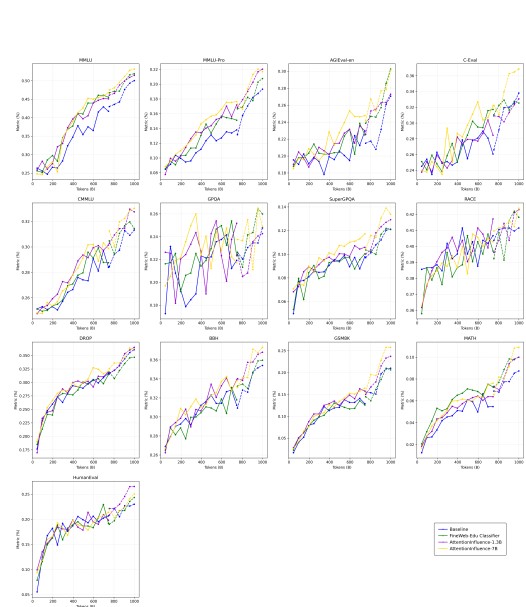

Figure 10: The performance evolution during pretraining on knowledge-intensive and reasoning-heavy benchmarks (i.e., MMLU, MMLU-Pro, AGIEval-en, C-Eval, CMMLU, GPQA, SuperGPQA, RACE, DROP, BBH, GSM8K, MATH, and HumanEval). The first 746B tokens correspond to the standard pretraining phase (solid lines), followed by 254B tokens under learning rate annealing (dashed lines). Curves with the same color (solid and dashed) indicate training on the same dataset.

| Model | #Tokens | Avg. | Metrics | | | | | | | |
|---|---|---|---|---|---|---|---|---|---|---|
| | | | ARC-C | ARC-E | ARC(C+E) | Wino. | Hella. | CSQA | OpenBookQA | PIQA |
| AttentionInfluence-1.3B w/o LRD | 495B | 37.39 | 52.13 | 80.35 | 66.24 | 65.19 | 71.40 | 44.39 | 45.20 | 77.09 |
| | | | TriviaQA | MMLU | MMLU-Pro | AGIEval-en | GPQA | SuperGPQA | RACE | DROP |
| | | | 43.43 | 39.72 | 14.38 | 21.51 | 24.26 | 10.04 | 39.04 | 29.88 |
| | | | BBH | GSM8K | MATH | HumanEval | C-Eval | CMMLU | | |
| | | | 33.45 | 12.51 | 6.05 | 17.87 | 27.93 | 29.37 | | |
| | | | ARC-C | ARC-E | ARC(C+E) | Wino. | Hella. | CSQA | OpenBookQA | PIQA |
| AttentionInfluence-7B w/o LRD | 495B | 37.96 | 51.28 | 79.55 | 65.42 | 65.04 | 71.29 | 52.42 | 44.60 | 78.18 |
| | | | TriviaQA | MMLU | MMLU-Pro | AGIEval-en | GPQA | SuperGPQA | RACE | DROP |
| | | | 44.49 | 42.64 | 15.66 | 22.74 | 21.22 | 10.73 | 38.28 | 29.94 |
| | | | BBH | GSM8K | MATH | HumanEval | C-Eval | CMMLU | | |
| | | | 32.25 | 13.42 | 6.05 | 18.63 | 29.72 | 29.12 | | |
| | | | ARC-C | ARC-E | ARC(C+E) | Wino. | Hella. | CSQA | OpenBookQA | PIQA |
| AttentionInfluence-1.3B w/o LRD | 746B | 39.32 | 56.66 | 82.03 | 69.35 | 71.90 | 65.43 | 53.48 | 43.60 | 77.58 |
| | | | TriviaQA | MMLU | MMLU-Pro | AGIEval-en | GPQA | SuperGPQA | RACE | DROP |
| | | | 45.68 | 45.10 | 17.19 | 22.99 | 25.18 | 10.59 | 41.72 | 32.03 |
| | | | BBH | GSM8K | MATH | HumanEval | C-Eval | CMMLU | | |
| | | | 33.89 | 15.77 | 6.38 | 19.85 | 28.45 | 30.15 | | |
| | | | ARC-C | ARC-E | ARC(C+E) | Wino. | Hella. | CSQA | OpenBookQA | PIQA |
| AttentionInfluence-7B w/o LRD | 746B | 39.85 | 55.80 | 83.25 | 69.53 | 64.33 | 71.94 | 56.18 | 44.40 | 78.51 |
| | | | TriviaQA | MMLU | MMLU-Pro | AGIEval-en | GPQA | SuperGPQA | RACE | DROP |
| | | | 46.14 | 46.77 | 17.64 | 24.77 | 21.73 | 11.72 | 40.29 | 32.09 |
| | | | BBH | GSM8K | MATH | HumanEval | C-Eval | CMMLU | | |
| | | | 33.81 | 16.45 | 7.62 | 21.40 | 32.17 | 29.89 | | |
| | | | ARC-C | ARC-E | ARC(C+E) | Wino. | Hella. | CSQA | OpenBookQA | PIQA |
| AttentionInfluence-1.3B w/ LRD | 1T | 43.16 | 59.98 | 84.26 | 72.12 | 68.03 | 75.49 | 61.59 | 46.60 | 79.54 |
| | | | TriviaQA | MMLU | MMLU-Pro | AGIEval-en | GPQA | SuperGPQA | RACE | DROP |
| | | | 51.20 | 51.48 | 22.03 | 27.30 | 24.26 | 12.92 | 42.30 | 36.52 |
| | | | BBH | GSM8K | MATH | HumanEval | C-Eval | CMMLU | | |
| | | | 36.80 | 23.73 | 10.00 | 26.55 | 33.06 | 32.75 | | |
| | | | ARC-C | ARC-E | ARC(C+E) | Wino. | Hella. | CSQA | OpenBookQA | PIQA |
| AttentionInfluence-7B w/ LRD | 1T | 43.59 | 56.31 | 84.05 | 70.18 | 67.48 | 75.24 | 62.90 | 47.00 | 79.76 |
| | | | TriviaQA | MMLU | MMLU-Pro | AGIEval-en | GPQA | SuperGPQA | RACE | DROP |
| | | | 51.68 | 53.18 | 21.70 | 30.18 | 24.87 | 13.39 | 42.39 | 36.25 |
| | | | BBH | GSM8K | MATH | HumanEval | C-Eval | CMMLU | | |
| | | | 37.32 | 25.78 | 10.90 | 25.06 | 36.85 | 33.04 | | |

Table 8: The ablation results on various benchmarks. The LRD denotes learning rate decay.

## J  LLM-AS-A-JUDGE EXPERIMENT DETAILS

We use GPT-4o to evaluate the performance of different data selection methods on the FineWeb-Edu-dedup domain. On the one hand, since most of the data in FineWeb-Edu-dedup is related to education, we aim for the selected high-quality data to be highly relevant to this domain. Therefore, we design an Education Score based on whether the selected sample content is education-related. On the other hand, we want the selected samples to contain more complex, reasoning-intensive knowledge. Based on this criterion, we design a Reasoning Score.

In summary, we use the following prompt to instruct GPT-4o score the selected samples:

LLM-As-A-Judge

PROMPT:

Given a piece of text: **<Selected Sample>**. Determine whether the text has educational value. If it does, respond with 1; if not, respond with 0. Then, determine whether the text is reasoning-intensive — that is, whether it contains explicit or implicit logical reasoning chains. If it does, respond with 1; if not, respond with 0. Respond in the following format:

```
\#\#Educational Value Score
<educational value score>

\#\#Reasoning Intensive Score
<reasoning intensive score>
```

Although GPT-4o can also be used for scoring pretraining data, different domains require specially designed prompts. Moreover, the computational cost of using GPT-4o for scoring is very high, whereas AttentionInfluence-1.3B has a much lower computational overhead.

## K  DETAILS OF CLUSTERING

We obtain document embeddings using Sentence-BERT (Reimers and Gurevych, 2019) and apply K-means clustering with $k = 100$. For each cluster, we sample representative documents near the

Table 9: The full results on various benchmarks. The LRD denotes learning rate decay.

**Baseline w/o LRD — #Tokens: 495B — Avg.: 36.39**

| ARC-C | ARC-E | ARC(C+E) | Wino. | Hella. | CSQA | OpenBookQA | PIQA |
|---|---|---|---|---|---|---|---|
| 54.35 | 81.44 | 67.89 | 64.40 | 71.21 | 32.19 | 46.20 | 78.02 |
| **TriviaQA** | **MMLU** | **MMLU-Pro** | **AGIEval-en** | **GPQA** | **SuperGPQA** | **RACE** | **DROP** |
| 43.74 | 35.44 | 13.12 | 20.59 | 22.23 | 9.44 | 39.52 | 28.93 |
| **BBH** | **GSM8K** | **MATH** | **HumanEval** | **C-Eval** | **CMMLU** | | |
| 32.29 | 12.05 | 6.08 | 19.94 | 25.48 | 27.42 | | |

**PPL filter w/o LRD — #Tokens: 495B — Avg.: 36.54**

| ARC-C | ARC-E | ARC(C+E) | Wino. | Hella. | CSQA | OpenBookQA | PIQA |
|---|---|---|---|---|---|---|---|
| 53.07 | 80.35 | 66.71 | 65.51 | 70.73 | 39.97 | 44.40 | 78.40 |
| **TriviaQA** | **MMLU** | **MMLU-Pro** | **AGIEval-en** | **GPQA** | **SuperGPQA** | **RACE** | **DROP** |
| 43.69 | 39.53 | 13.33 | 20.80 | 22.30 | 9.20 | 39.43 | 27.71 |
| **BBH** | **GSM8K** | **MATH** | **HumanEval** | **C-Eval** | **CMMLU** | | |
| 29.50 | 9.10 | 5.16 | 20.27 | 28.10 | 26.70 | | |

**Scaling Filter w/o LRD — #Tokens: 495B — Avg.: 36.81**

| ARC-C | ARC-E | ARC(C+E) | Wino. | Hella. | CSQA | OpenBookQA | PIQA |
|---|---|---|---|---|---|---|---|
| 52.65 | 81.31 | 66.98 | 63.54 | 70.43 | 40.62 | 42.80 | 77.48 |
| **TriviaQA** | **MMLU** | **MMLU-Pro** | **AGIEval-en** | **GPQA** | **SuperGPQA** | **RACE** | **DROP** |
| 44.05 | 39.37 | 13.81 | 21.20 | 24.90 | 9.63 | 39.71 | 28.69 |
| **BBH** | **GSM8K** | **MATH** | **HumanEval** | **C-Eval** | **CMMLU** | | |
| 30.60 | 11.60 | 5.80 | 18.75 | 28.50 | 27.60 | | |

**FineWeb-Edu Classifier w/o LRD — #Tokens: 495B — Avg.: 37.44**

| ARC-C | ARC-E | ARC(C+E) | Wino. | Hella. | CSQA | OpenBookQA | PIQA |
|---|---|---|---|---|---|---|---|
| 54.35 | 81.73 | 68.04 | 64.96 | 70.34 | 46.60 | 44.00 | 77.58 |
| **TriviaQA** | **MMLU** | **MMLU-Pro** | **AGIEval-en** | **GPQA** | **SuperGPQA** | **RACE** | **DROP** |
| 43.17 | 41.00 | 13.36 | 20.46 | 22.94 | 9.36 | 40.67 | 30.08 |
| **BBH** | **GSM8K** | **MATH** | **HumanEval** | **C-Eval** | **CMMLU** | | |
| 30.94 | 12.51 | 7.10 | 18.66 | 28.45 | 27.97 | | |

**AttentionInfluence-1.3B w/o LRD — #Tokens: 495B — Avg.: 37.39**

| ARC-C | ARC-E | ARC(C+E) | Wino. | Hella. | CSQA | OpenBookQA | PIQA |
|---|---|---|---|---|---|---|---|
| 52.13 | 80.35 | 66.24 | 65.19 | 71.40 | 44.39 | 45.20 | 77.09 |
| **TriviaQA** | **MMLU** | **MMLU-Pro** | **AGIEval-en** | **GPQA** | **SuperGPQA** | **RACE** | **DROP** |
| 43.43 | 39.72 | 14.38 | 21.51 | 24.26 | 10.04 | 39.04 | 29.88 |
| **BBH** | **GSM8K** | **MATH** | **HumanEval** | **C-Eval** | **CMMLU** | | |
| 33.45 | 12.51 | 6.05 | 17.87 | 27.93 | 29.37 | | |

**AttentionInfluence-7B w/o LRD — #Tokens: 495B — Avg.: 37.96**

| ARC-C | ARC-E | ARC(C+E) | Wino. | Hella. | CSQA | OpenBookQA | PIQA |
|---|---|---|---|---|---|---|---|
| 51.28 | 79.55 | 65.42 | 65.04 | 71.29 | 52.42 | 44.60 | 78.18 |
| **TriviaQA** | **MMLU** | **MMLU-Pro** | **AGIEval-en** | **GPQA** | **SuperGPQA** | **RACE** | **DROP** |
| 44.49 | 42.64 | 15.66 | 22.74 | 21.22 | 10.73 | 38.28 | 29.94 |
| **BBH** | **GSM8K** | **MATH** | **HumanEval** | **C-Eval** | **CMMLU** | | |
| 32.25 | 13.42 | 6.05 | 18.63 | 29.72 | 29.12 | | |

**Baseline w/o LRD — #Tokens: 746B — Avg.: 38.21**

| ARC-C | ARC-E | ARC(C+E) | Wino. | Hella. | CSQA | OpenBookQA | PIQA |
|---|---|---|---|---|---|---|---|
| 55.89 | 81.69 | 68.79 | 66.22 | 71.79 | 49.14 | 45.40 | 79.27 |
| **TriviaQA** | **MMLU** | **MMLU-Pro** | **AGIEval-en** | **GPQA** | **SuperGPQA** | **RACE** | **DROP** |
| 45.57 | 41.76 | 13.80 | 22.92 | 21.93 | 9.78 | 40.67 | 31.71 |
| **BBH** | **GSM8K** | **MATH** | **HumanEval** | **C-Eval** | **CMMLU** | | |
| 31.23 | 12.89 | 5.48 | 20.70 | 26.08 | 28.40 | | |

**FineWeb-Edu Classifier w/o LRD — #Tokens: 746B — Avg.: 38.77**

| ARC-C | ARC-E | ARC(C+E) | Wino. | Hella. | CSQA | OpenBookQA | PIQA |
|---|---|---|---|---|---|---|---|
| 55.12 | 82.74 | 68.93 | 64.33 | 71.78 | 53.15 | 45.00 | 78.84 |
| **TriviaQA** | **MMLU** | **MMLU-Pro** | **AGIEval-en** | **GPQA** | **SuperGPQA** | **RACE** | **DROP** |
| 45.17 | 45.56 | 15.12 | 22.48 | 22.34 | 10.04 | 39.71 | 31.47 |
| **BBH** | **GSM8K** | **MATH** | **HumanEval** | **C-Eval** | **CMMLU** | | |
| 31.79 | 12.59 | 7.28 | 19.21 | 31.72 | 28.76 | | |

**AttentionInfluence-1.3B w/o LRD — #Tokens: 746B — Avg.: 39.32**

| ARC-C | ARC-E | ARC(C+E) | Wino. | Hella. | CSQA | OpenBookQA | PIQA |
|---|---|---|---|---|---|---|---|
| 56.66 | 82.03 | 69.35 | 65.43 | 71.90 | 53.48 | 43.60 | 77.58 |
| **TriviaQA** | **MMLU** | **MMLU-Pro** | **AGIEval-en** | **GPQA** | **SuperGPQA** | **RACE** | **DROP** |
| 45.68 | 45.10 | 17.19 | 22.99 | 25.18 | 10.59 | 41.72 | 32.03 |
| **BBH** | **GSM8K** | **MATH** | **HumanEval** | **C-Eval** | **CMMLU** | | |
| 33.89 | 15.77 | 6.38 | 19.85 | 28.45 | 30.15 | | |

**AttentionInfluence-7B w/o LRD — #Tokens: 746B — Avg.: 39.85**

| ARC-C | ARC-E | ARC(C+E) | Wino. | Hella. | CSQA | OpenBookQA | PIQA |
|---|---|---|---|---|---|---|---|
| 55.80 | 83.25 | 69.53 | 64.33 | 71.94 | 56.18 | 44.40 | 78.51 |
| **TriviaQA** | **MMLU** | **MMLU-Pro** | **AGIEval-en** | **GPQA** | **SuperGPQA** | **RACE** | **DROP** |
| 46.14 | 46.77 | 17.64 | 24.77 | 21.73 | 11.72 | 40.29 | 32.09 |
| **BBH** | **GSM8K** | **MATH** | **HumanEval** | **C-Eval** | **CMMLU** | | |
| 33.81 | 16.45 | 7.62 | 21.40 | 32.17 | 29.89 | | |

**Baseline w/ LRD — #Tokens: 1T — Avg.: 42.46**

| ARC-C | ARC-E | ARC(C+E) | Wino. | Hella. | CSQA | OpenBookQA | PIQA |
|---|---|---|---|---|---|---|---|
| 58.79 | 83.92 | 71.36 | 70.24 | 75.63 | 59.62 | 48.00 | 80.63 |
| **TriviaQA** | **MMLU** | **MMLU-Pro** | **AGIEval-en** | **GPQA** | **SuperGPQA** | **RACE** | **DROP** |
| 51.07 | 50.05 | 19.32 | 27.06 | 24.77 | 12.10 | 41.15 | 36.09 |
| **BBH** | **GSM8K** | **MATH** | **HumanEval** | **C-Eval** | **CMMLU** | | |
| 35.42 | 21.00 | 8.74 | 23.02 | 33.80 | 31.33 | | |

**FineWeb-Edu Classifier w/ LRD — #Tokens: 1T — Avg.: 42.66**

| ARC-C | ARC-E | ARC(C+E) | Wino. | Hella. | CSQA | OpenBookQA | PIQA |
|---|---|---|---|---|---|---|---|
| 57.85 | 83.67 | 70.76 | 68.03 | 75.21 | 61.59 | 47.00 | 80.09 |
| **TriviaQA** | **MMLU** | **MMLU-Pro** | **AGIEval-en** | **GPQA** | **SuperGPQA** | **RACE** | **DROP** |
| 49.93 | 51.92 | 20.76 | 30.27 | 25.99 | 12.12 | 41.82 | 34.68 |
| **BBH** | **GSM8K** | **MATH** | **HumanEval** | **C-Eval** | **CMMLU** | | |
| 35.97 | 20.62 | 10.00 | 24.36 | 32.54 | 31.45 | | |

**AttentionInfluence-1.3B w/ LRD — #Tokens: 1T — Avg.: 43.16**

| ARC-C | ARC-E | ARC(C+E) | Wino. | Hella. | CSQA | OpenBookQA | PIQA |
|---|---|---|---|---|---|---|---|
| 59.98 | 84.26 | 72.12 | 68.03 | 75.49 | 61.59 | 46.60 | 79.54 |
| **TriviaQA** | **MMLU** | **MMLU-Pro** | **AGIEval-en** | **GPQA** | **SuperGPQA** | **RACE** | **DROP** |
| 51.20 | 51.48 | 22.03 | 27.30 | 24.26 | 12.92 | 42.30 | 36.52 |
| **BBH** | **GSM8K** | **MATH** | **HumanEval** | **C-Eval** | **CMMLU** | | |
| 36.80 | 23.73 | 10.00 | 26.55 | 33.06 | 32.75 | | |

**AttentionInfluence-7B w/ LRD — #Tokens: 1T — Avg.: 43.59**

| ARC-C | ARC-E | ARC(C+E) | Wino. | Hella. | CSQA | OpenBookQA | PIQA |
|---|---|---|---|---|---|---|---|
| 56.31 | 84.05 | 70.18 | 67.48 | 75.24 | 62.90 | 47.00 | 79.76 |
| **TriviaQA** | **MMLU** | **MMLU-Pro** | **AGIEval-en** | **GPQA** | **SuperGPQA** | **RACE** | **DROP** |
| 51.68 | 53.18 | 21.70 | 30.18 | 24.87 | 13.39 | 42.39 | 36.25 |
| **BBH** | **GSM8K** | **MATH** | **HumanEval** | **C-Eval** | **CMMLU** | | |
| 37.32 | 25.78 | 10.90 | 25.06 | 36.85 | 33.04 | | |

| Domain | FineWeb-Edu Classifier | | | AttentionInfluence | | |
|---|---|---|---|---|---|---|
| | Education Score | Reasoning Score | Token Len | Education Score | Reasoning Score | Token Len |
| FineWeb-Edu-dedup | 0.99 | 0.52 | 1610.12 | 0.99 | 0.49 | 1629.73 |
| Cosmopedia-V2 | 1.00 | 0.87 | 825.46 | 1.00 | 0.80 | 893.80 |
| Python-Edu | 0.98 | 0.76 | 414.15 | 0.98 | 0.87 | 820.71 |
| OpenWebMath | 0.99 | 0.52 | 1022.86 | 0.96 | 0.88 | 2255.57 |

Table 10: The quality score of the data selected by AttentionInfluence and FineWeb-Edu Classifier.

| Domain | 1.3B | | | 7B | | |
|---|---|---|---|---|---|---|
| | Education Score | Reasoning Score | Token Len | Education Score | Reasoning Score | Token Len |
| FineWeb-Edu-dedup | 0.99 | 0.49 | 1895.7 | 0.97 | 0.58 | 3488.8 |
| Cosmopedia-V2 | 1.0 | 0.80 | 2774.6 | 1.0 | 0.82 | 2984.1 |
| Python-Edu | 0.97 | 0.87 | 909.3 | 0.98 | 0.91 | 1657.2 |
| OpenWebMath | 0.96 | 0.88 | 2138.6 | 0.96 | 0.93 | 5550.4 |

Table 11: The quality score of the data selected by AttentionInfluence using 1.3B and 7B models.

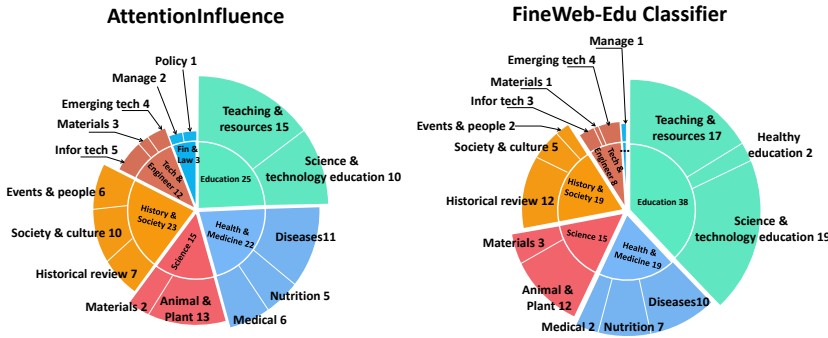

Figure 11: The statistics of clustering. The left is the clustering result of AttentionInfluence, the right part is that of FineWeb-Edu Classifier.

cluster center and use GPT-4o to generate descriptive fine-grained (i.e., secondary) category labels, such as *Education–Teaching & Resources*.

We manually group these secondary labels into six primary categories and report the number of samples falling into each high-level category for both selection methods, which is shown in Figure 11.

## L  CASE STUDY

In this section, we present the cases selected by FineWeb-Edu Classifier and AttentionInfluence-1.3B.

| Method | Ranking | Words |
|---|---|---|
| AttentionInfluence | 0%- 1% | frac, len, sklearn, append, pyplot, browser, pre, mathbf, 3d, employee, __init__ |
| | 1%- 10% | well, part, movement, children, appreciation, involve, remember, family growth, treatment, principles, business, b, long, work |
| | 10%- 50% | maximize, paintings, independence, therefore, expenses, regulatory, recall square, protocols, monitoring, integrity, consistent, channels, inspiring, width |
| | 50%- 100% | driver, flying, humble, fourier, smoother, longstanding, owl personnel, lawyers, entrenched, beach, brother, oils, wow, desk |
| FineWeb-Edu Classifier | 0%- 1% | dimensional, student, 3d, 19th, eco, anti israelite, bmatrix, voter, socio, linspace |
| | 1%- 10% | creative, based, would, sources, do, system, compared, someone studies, delve, true, turn, only, elements, ultimately |
| | 10%- 50% | argument, bright, rising, excessive, governments, friendships, complicated, discipline constitutes, hearing, consequences, institutional, match, meets, holocaust |
| | 50%- 100% | peek, manifest, reciprocity, obligations, toilet, customized, olive validity, enriching, profits, presentations, twelve, originating, arithmetic, nazi |

Table 12: The high-frequency words of different methods.

| Rank: top 0.24% (AttentionInfluence-1.3B) | Rank: top 0.74% (FineWebEdu Classifier) |
|---|---|
| Consider a board similar to the one below\\     7 8 9 10 \\     6 1 2 11\\     5 4 3   \\ However, imagine it as being infinite. A die is initially placed at 1 and can only move to the next consecutive number (e.g 1 to 2, 2 to 3...)  Prompts the user for a natural number N at least equal to 1, and outputs the numbers at the top, the front and the  right after the die has been moved to cell N.

Written by Benny Hwang 13/08/2017

import math
def move_right(Current_faces):
Top_old = Current_faces[0]
Right_old = Current_faces[2]
Bottom_old = Current_faces[3]
Left_old = Current_faces[5]

….

if __name__ == '__main__':
N = False
while N == False:
   …. | # Chapter 01
# Exercise 04
# Write a method to replace all spaces in a string with '%20'
# Pretty basic for Python
def main():
test_string = ""This is a test string""
print spaces(test_string)
def spaces_easy(input):
return input.replace(' ','%20')
if __name__ == ""__main__"":
main() |

Figure 12: The sample in Python-Edu domain ranked within the top 20% according to AttentionInfluence-1.3B (**left**) an FineWeb-Edu Classifier (**right**).

| Rank: top 0.24% (AttentionInfluence-1.3B) | Rank: top 0.74% (FineWebEdu Classifier) |
|---|---|
| "17Calculus - Vector Cross Product Application - Triple Scalar Product

17Calculus

The triple scalar product is a result of combining the dot product with thecross product. Some other names for the triple scalarproduct are scalar triple product, mixed product and box product.First, let's define what it is and then discuss a couple of properties.

Definition and Notation

If we have three vectors in space $$\vec{u} = u_x\hat{i}+u_y\hat{j}+u_z\hat{k}$$$$\vec{v} = v_x\hat{i}+v_y\hat{j}+v_z\hat{k}$$ and$$\vec{w} = w_x\hat{i}+w_y\hat{j}+w_z\hat{k}$$, then the triple scalar product is defined to be $$\vec{u} \cdot (\vec{v} \times \vec{w})$$ The calculation of this can be done as follows$$\vec{u} \cdot (\vec{v} \times \vec{w}) = \begin{vmatrix} u_x & u_y & u_z \\ v_x & v_y & v_z \\ w_x & w_y & w_z \end{vmatrix}$$Let's look at where this comes from.

Theorem: Triple Scalar Product

If we have three vectors in space,$$\vec{u} = u_x\hat{i}+u_y\hat{j}+u_z\hat{k}$$, $$\vec{v} = v_x\hat{i}+v_y\hat{j}+v_z\hat{k}$$ and $$\vec{w} = w_x\hat{i}+w_y\hat{j}+w_z\hat{k}$$, ……. | # Compressibility

(Redirected from Incompressible)
"Incompressible" redirects here. For the property of vector fields, see Solenoidal vector field. For the topological property, see Incompressiblesurface.

In thermodynamics and fluid mechanics, compressibility is a measure ofthe relative volume change of a fluid or solid as a response to a pressure(or mean stress) change.

$\beta=-\frac{1}{V}\frac{\partial V}{\partial p}$where V is volume and p is pressure.

## Definition
…

## Fluid dynamics

The degree of compressibility of a fluid has strong implications for its dynamics. Most notably, the propagation of sound is dependent on the compressibility of the medium.

### Aeronautical dynamics

Compressibility is an important factor in aerodynamics.  …. |

Figure 13: The sample in OpenWebMath domain ranked within the top 20% according to AttentionInfluence-1.3B (**left**) and FineWeb-Edu Classifier (**right**).

| Sample1 (Tag: Health & Medicine - Health Guidelines & Nutrition) | Sample2 (Tag: Technology and Engineering - Information Technology) |
|---|---|
| Type 2 diabetes is a chronic illness costing over \$300 billion per year in the United States with an estimated 100 million individuals with diabetes or pre-diabetes. Complications due to diabetes place individuals at increased risk for heart attack, stroke, amputations, blindness, kidney failure, disability, and early death. Education has been shown to be effective in improving health behaviors that decrease complications due to diabetes. Common risk factors for development of diabetes are modifiable behaviors such as sedentary lifestyle and obesity.A peer-led approach to diabetes education has the potential to overcome multiple barriers to receiving education. Peer-led diabetes education can provide education at low or no cost in communities where individuals feel welcomed and travel is minimized. Diabetes education has the potential to decrease disability, early death, and the economic costs of diabetes. | Bitcoin mining is a process of verifying transactions and recording them on the blockchain ledger. The blockchain is a decentralized public ledger that keeps a record of all Bitcoin transactions. Mining involves solving complex mathematical problems using specialized software and hardware. Explore qumasai.io for further information. |
|  | The Bitcoin network rewards miners for successfully verifying transactions by giving them newly created Bitcoins. The mining process involves adding a new block of transactions to the blockchain every 10 minutes. Miners compete against each other to add the next block to the chain. |
| The purpose of this study was to determine if peer-led sessions on diabetes self-management impacted health behaviors, empowerment, and knowledge of diabetes. Four topic-driven educational sessions were provided for participants in Northeast Arkansas who had either a diagnosis of pre-diabetes or diabetes. Pre and post-questionnaires were used to assess changes in knowledge using the Revised Diabetes Knowledge Test, empowerment using the Diabetes Empowerment Scale - Short Form, and health behaviors. | To participate in Bitcoin mining, one needs to have a powerful hardware setup and specialized mining software. The hardware required is called an ASIC miner, which is specially designed to solve the mathematical problems required to add a block to the blockchain. |
| A statistically significant difference was found in the empowerment scale with an increase in mean scores from 31.23 to 36.04. A paired samples t-test found a statistically significant difference in scores on Diabetes Knowledge Test, (t (25) = −2.54, p < .05). Significant changes in health behaviors were found for knowledge of A1C levels, the frequency of foot exams, and days of exercise per week.Focus groups following intervention provided qualitative results indicating satisfaction with the peer-led model. In order to implement peer-led education, there is a need to develop improved strategies for recruitment. A peer-led model for diabetes education has potential to provide needed education. | The Bitcoin network is designed to gradually decrease the mining reward over time. As the number of Bitcoins in circulation increases, the mining reward decreases. This is done to maintain the scarcity and value of Bitcoin.Bitcoin mining requires a significant amount of energy, which has led to concerns about its environmental impact. However, many miners are taking steps to use renewable energy sources to power their mining operations.In summary, Bitcoin mining is a competitive process that involves verifying transactions and adding them to the blockchain ledger. It requires specialized hardware and software and rewards miners with newly created Bitcoins. Although it consumes a significant amount of energy, advances in renewable energy are making Bitcoin mining more sustainable..What exactly is Bitcoin mining? |
| \|Commitee:\|\|Guffey, James S., Hall, John, Nichols, Joseph, Nix, Elizabeth\|\|School:\|\|Arkansas State University\|\|School Location:\|\|United States -- Arkansas\|\|Source:\|\|DAI-A 80/09(E), Dissertation Abstracts International\|\|Subjects:\|\|Educational leadership, Public Health Education, Nutrition\|\|Keywords:\|\|Community, Diabetes, Education, Peer-led\| | Bitcoin mining is the process of adding new transactions to the blockchain and verifying them. It's done by solving complex mathematical problems and recording those transactions on a public ledger known as the blockchain. The miners who successfully solve these problems are rewarded with newly generated bitcoins.The mining process involves many miners around the world competing to solve these problems, and the first one to do so earns the reward, which is currently 6.25 bitcoins. This reward is then divided among the miners who participated in the process.But mining bitcoin requires a lot of computing power, which means it requires a lot of energy. In fact, according to the Cambridge Bitcoin Electricity Consumption Index, bitcoin mining now consumes as much energy as Switzerland, a country with a population of 8 million.Despite its energy consumption, Bitcoin mining is essential to the functioning of the currency. Without mining, there would be no way to ensure the integrity of the transactions, and the decentralized nature of the currency would be undermined.In recent years, some critics have raised concerns about the environmental impact of Bitcoin mining. However, efforts are being made to reduce the energy consumption associated with the process, … |
| Copyright in each Dissertation and Thesis is retained by the author. All Rights ReservedThe supplemental file or files you are about to download were provided to ProQuest by the author as part of adissertation or thesis. The supplemental files are provided "AS IS" without warranty. ProQuest is not responsible for thecontent, format or impact on the supplemental file(s) on our system. in some cases, the file type may be unknown ormay be a .exe file. We recommend caution as you open such files. |  |

Figure 14: The samples of a clustering in data in the Cosmopedia-V2 domain ranked within 20% according to AttentionInfluence.

## M CLUSTERING CASE

As shown in Figure 14, we present the two clustering cases in the Cosmopedia-V2 domain.

| 1.3B (Top 0.10%) | 1.3B (Top 97.95% ) |
| --- | --- |
| ## Modeling Dynamic Systems in Python\n\nIn this section, we will explore how to model dynamic systems using Python. We will focus on a specific example involving the equations of motion for an aircraft, but the concepts and techniques we cover will be applicable to a wide range of dynamic systems.\n\n### Equations of Motion\n\nThe equations of motion for an aircraft can be quite complex, as they involve multiple coordinate systems and take into account various forces and moments acting on the aircraft. However, we can simplify the problem by considering a specific set of equations known as the "flat Earth equations." These equations assume that the Earth is flat and non-rotating, which is a reasonable approximation for many applications.\n\nThe flat Earth equations can be written in the following form:\n```python\not = (q * sin(phi) + r * cos(phi)) / cos(theta)\n```\nwhere `ot` is the "out-of-track" error, which represents the lateral deviation of the aircraft from its intended course. The variables `q`, `r`, `phi`, and `theta` are related to the aircraft\'s angular rates and orientation.\n\n### Moment Equations\n\nThe moment equations describe how the angular rates of the aircraft change over time. These equations take into account the moments generated by the aircraft\'s control surfaces, as well as any external disturbances such as wind gusts.\n\nThe moment equations can be written in the following form:\n```python\np_dot = (j_xz * (j_x - j_y + j_z) * p * q - (j_z * (j_z - j_y) + j_xz ** 2) * q * r + j_z * roll + j_xz * yaw )/ gamma\nq_dot = ((j_z - j_x) * p * r - j_xz * (p ** 2 - r ** 2) + pitch) / j_y\nr_dot = (((j_x - j_y) * j_x + j_xz ** 2) * p * q - j_xz * (j_x - j_y + j_z) * q * r + j_xz * roll + j_x * yaw )/ gamma\n```\nwhere `p_dot`, `q_dot`, and `r_dot` are the time derivatives of the angular rates, `j_x`, `j_y`, and `j_z` are … | I remember watching this indie film last year that really got me thinking about the way society treats certain racial and ethnic groups. It was called "Beyond Skin Deep" and told the story of a young African American woman named Tasha who moves to a small, predominantly white town in the Midwest. Throughout the movie, we see how Tasha faces subtle (and not-so-subtle) racism from her neighbors, coworkers, and even some friends. But what struck me most were the scenes showing how she struggled to fit in and find a sense of belonging in a community that seemed to reject her at every turn. One scene in particular has stuck with me. Tasha is at a local bar with some colleagues after work, trying to make conversation and connect with them. But instead of engaging with her, they talk over her, ignore her contributions to the conversation, and eventually leave without inviting her along. As she watches them go, tears well up in her eyes and she looks around the now-empty bar, feeling completely alone and isolated. What made this film so powerful, in my opinion, was the way it used depictions of race and ethnicity to shed light on broader societal frustrations. By focusing on one character\'s experience, it highlighted the systemic issues that many people of color face on a daily basis - things like microaggressions, implicit bias, and exclusion. But just when you think you know where the story is going, there\'s an unexpected plot twist. It turns out that Tasha isn\'t actually African American - she\'s Middle Eastern, but had been passing as black because she felt more accepted in that community than in her own. This revelation forces us to reevaluate everything we thought we knew about Tasha\'s struggles, and challenges us to consider the ways in which our assumptions and prejudices can blind us to the true complexities …… |
| 7B (Top 0.10%) | 7B (Top 97.95%) |
| ## Understanding Dictionaries and Lists in Python\n\nPython is a powerful programming language that allows us to work with different types of data. In this unit, we will explore two essential data structures: dictionaries and lists. We will also learn how to manipulate and analyze data using these structures.\n\n### Dictionaries\n\nA dictionary in Python is a collection of key-value pairs. It is an unordered collection, meaning that the items do not have a specific order. Each key-value pair is called an item. The syntax for creating a dictionary is as follows:\n```python\nmy_dict = {\n   "key1": "value1",\n   "key2": "value2",\n   "key3": "value3"\n}\n```\n\nYou can access the values in a dictionary using their corresponding keys:\n\n```python\nprint(my_dict["key1"]) # Output: "value1"\n```\n\n### Lists\n\nA list in Python is an ordered collection of items. It is similar to an array in other programming languages. The syntax for creating a list is as follows:\n\n```python\nmy_list = ["item1", "item2", "item3"]\n```\n\nYou can access the items in a list using their index, which starts at 0:\n\n```python\nprint(my_list[0]) # Output: "item1"\n```\n\n## Analyzing Data with Dictionaries and Lists\n\nNow that we have a basic understanding of dictionaries and lists, let\'s explore how we can use them to analyze data. We will use a code snippet from a Python tutorial as an example.\n\n### The Code Snippet\n\nHere is the code snippet we will be analyzing:\n\n```python\nresult[track.name] = {\n    "cues": firstK, # Cues candidates\n    "cuesFeature": {\n      features[j]: len([1 for t in signal.times if t in firstK]) / len(firstK) if len(firstK) else 0\n      for j, signal in enumerate(peakSignals)\n   },\n}\n```\nif any(gttracks):\n    gtCues += gttracks[i].features["boundaries"]\n    result[track.name]["gtCues"] = gttracks[i].features["boundaries"] # Cues annotated\n    result[track.name]["gtCuesFeature"] = {\n        features[j]: len([\ … | In today's digital age, businesses rely heavily on complex computer networks to connect their operations, communicate with clients, and store vast amounts of data. At the heart of these networks lies the work of skilled networking professionals who design, implement, and maintain these critical systems. If you are interested in pursuing a career in this field, obtaining a CCNA (Cisco Certified Network Associate) certification can serve as an excellent starting point. In particular, gaining expertise in CCAr (Cisco Certified Architect) architecture can set you apart as a true leader in network design and strategy. Before diving into the specifics of CCAr architecture, it's essential to understand the foundational principles that underpin all networking technologies. At its core, networking involves connecting multiple devices—such as computers, servers, and smartphones—to enable communication and resource sharing. To accomplish this goal, networks employ various layers of hardware and software components working together to transmit information between nodes efficiently and securely. These layers follow well-defined standards and protocols, ensuring seamless interoperability across different vendors and platforms. As a leading provider of networking equipment and solutions, Cisco has established itself as a dominant force within the industry. With a diverse range of products catering to organizations of all sizes, Cisco offers numerous certifications designed to validate the skills and knowledge of networking professionals at every stage of their careers. Among them, the CCNA stands out as an ideal entry point for those new to the field, providing a solid foundation in networking fundamentals while also serving as a stepping stone toward more advanced credentials like the CCAr. Obtaining a CCNA certification requires passing a single exam, known as ….. |

Figure 15: The cases of AttentionInfluence in Cosmopeida-V2 domain.

# N    CASES OF ATTENTIONINFLUENCE BASED ON 1.3B AND 7B MODELS

As shown in Figure 15, Figure 16, Figure 17, and Figure 18, we present some cases with different score levels.

| 1.3B (Top 0.10%) | 1.3B (Top 97.95% ) |
|---|---|
| Excel is a popular tool for data analysis, and its usage has increased significantly in recent years. It provides numerous features that make managing data easier. One such feature is the 'Save As' function that helps users create a copy of an existing Excel file with a new name and file format. In this article, we will discuss the 'Save As' function and the keyboard shortcut used for it.\nWhat is the 'Save As' function in Excel?\nThe 'Save As' function in Excel allows users to create a copy of an existing file and save it with a new name or file format. This function is useful when you want to make a copy of an Excel file as a backup, create a new version of the file, or save the file in a different format that is compatible with other applications or systems.\nWhy is the 'Save As' function important?\nThe 'Save As' function is essential because it helps users avoid overwriting their original files accidentally. When you save an Excel file using the 'Save As' function, a new copy of the file is created, and the original file remains unchanged. This way, you can always revert to the original file if necessary.\nWhat is the keyboard shortcut for the 'Save As' function in Excel?\nThe keyboard shortcut for the 'Save As' function in Excel is 'F12'. Pressing the 'F12' key brings up the 'Save As' dialog box, where you can choose the location, name, and file format for the new copy of the file.\nHow to use the 'Save As' function using the keyboard shortcut?\nUsing the 'Save As' function using the keyboard shortcut is easy. Follow the steps below:\n- Open the Excel file you want to save as a new copy\n- Press 'F12' on your keyboard\n- The 'Save As' dialog box will appear\n- Choose the location where you want to save the new copy of the file\n- Enter a new name for the file in the 'File name' field\n- Select the file format you want to use from the 'Save as type' dropdown menu\n- Click the 'Save' button\nWhat are the benefits of using the keyboard shortcut….. | - Nano Fish Limnophila hippuridoides is originally from Asia and the stalks grow to be 20-50 cm high and 6-10 cm wide – often with beautiful outwards crooked shoot tips. A simple plant, able to adjust to various conditions. The leaves are green with a red-violet underside, and the whole leaf turns red-violet under ideal growth conditions. A vigorously growing plant that willingly creates new, solid shoots from the base. Thinning of the oldest and longest shoots is recommended, in order to make room for such new shoots. Replant the cut-offs, they will soon grow new roots. If either stem or leaves are damaged, a strong scent is emitted. Growth rate: Medium Height: 20 - 30+ Light demand: Medium $CO_2$ : Low |

| 7B (Top 0.10%) | 7B (Top 97.95%) |
|---|---|
| Understanding the Three Common Causes of Sensor Failure\nIn today's technologically advanced world, sensors play a crucial role in various industries, from automotive to healthcare. These devices are designed to detect and measure physical properties, enabling machines and systems to operate efficiently. However, like any other piece of technology, sensors are not immune to failure. Understanding the common causes behind sensor failure is essential for businesses and individuals relying on these devices to ensure smooth operations and prevent costly disruptions.\nOne of the primary causes of sensor failure is environmental factors. Sensors are often exposed to harsh conditions, such as extreme temperatures, humidity, or corrosive substances. Over time, these factors can degrade the sensor's components, leading to inaccurate readings or complete malfunction. For instance, in industrial settings where sensors are exposed to high temperatures or corrosive chemicals, the lifespan of the sensor may be significantly reduced. It is crucial to select sensors that are specifically designed to withstand the environmental conditions they will be exposed to, ensuring their longevity and reliability.\nAnother common cause of sensor failure is mechanical stress. Sensors are often subjected to physical forces, such as vibrations, shocks, or excessive pressure. These external forces can damage the delicate internal components of the sensor, resulting in inaccurate measurements or complete failure. For example, in automotive applications, sensors may be exposed to constant vibrations or sudden impacts, which can lead to premature failure if not properly protected. Employing appropriate mounting techniques and using protective measures, such as shock absorbers or vibration dampeners, can help mitigate the risk of mechanical stress-induced sensor failure.\nElectrical issues also contribute significantly to sensor failure. Power surges, voltage spikes, ….. | An eye-opening look at the life and legacy of Jackie Robinson, the man who broke the color barrier in Major League Baseball and became an American hero. Baseball, basketball, football — no matter the game, Jackie Robinson excelled. His talents would have easily landed another man a career in pro sports, but such opportunities were closed to athletes like Jackie for one reason: his skin was the wrong color. Settling for playing baseball in the Negro Leagues, Jackie chafed at the inability to prove himself where it mattered most: the major leagues. Then in 1946, Branch Rickey, manager of the Brooklyn Dodgers, recruited Jackie Robinson. Jackie faced cruel and sometimes violent hatred and discrimination, but he proved himself again and again, exhibiting courage, determination, restraint, and a phenomenal ability to play the game. In this compelling biography, award-winning author Doreen Rappaport chronicles the extraordinary life of Jackie Robinson and how his achievements won over — and changed — a segregated nation. Potentially Sensitive Areas: Violence, Racism and racist language Booklist (September 1, 2017 (Vol. 114, No. 1)) Grades 5-7. Early on, young Jackie Robinson was taught to fight back when faced with racial slurs and prejudice, and he did, first as one of the few black kids in his neighborhood and later as one of the few black officers on his army base. But those injustices and the indignities he endured while playing for Negro league baseball were dwarfed by the hostility shown by many white players and fans when he broke the color barrier in Major League Baseball. While children's ….. |

Figure 16: The cases of AttentionInfluence in FineWeb-Edu-dedup domain.

| 1.3B (Top 0.10%) | 1.3B (Top 97.95% ) |
|---|---|
| The Associative Property of Addition is one of four basic properties that students will learn in early addition lessons and use later in multiplication and pre-algebra. Remembering the formula for commutative property of addition is a + b = b + a and you are good to go! The commutative property is a fundamental building block of math, but it only works for addition and multiplication. By non-commutative, we mean the switching of the order will give different results. Example 1: 2 + 4 = 4 + 2 = 6 . What is the Commutative Property? The mathematical operations, subtraction and division are the two non-commutative operations. You can find them all at the bottom of this page. The commutative property for any two numbers, X and Y, is X # Y = Y # X where # can stand for addition or multiplication. The commutative property of addition essentially states that no matter what order the addends are in within a particular number sentence, the sums will be the same. The product of any number and 0 is 0 For example: 874 × 0 = 0 Identity Property of Addition & … Subtraction (Not Commutative) Subtraction is probably an example that you know, intuitively, is not commutative . 16y + 0 = 16y Associate Property of Addition Zero Property of Multiplication Commutative Property of Addition Identity Property of Addition 2. d • r = r • d Commutative Property of Multiplication Identity Property of . This rule just says that, when you are doing addition, it doesn\'t matter which order the numbers are in. Just enter the inputs, the commutative property of addition calculator will update you the result. Addition and multiplication are both commutative. The properties include the commutative, identity, and distributive properties--all of which I cover in other math lessons. The commutative property of addition also applies to variables similarly. Commutative Property Of Addition: …… | Article  Impact Of Fading Correlation And Unequal Branch Gains On The Capacity Of Diversity Systems Dept. of Electr. Eng., California Inst. of Technol., Pasadena, CA Vehicular Technology Conference, 1988, IEEE 38th 11/2001; DOI:10.1109/VETEC.1999.778436 In proceeding of: Vehicular Technology Conference, 1999 IEEE 49th, Volume: 3 Source: IEEE Xplore ABSTRACT We investigate the effect of fading correlation and branch gain imbalance on the Shannon capacity of diversity systems in conjunction with adaptive transmission techniques. This capacity provides the theoretical upper bound for the spectral efficiency of adaptive transmission schemes. We obtain closed-form expressions for this capacity for Rayleigh fading channels under four adaptation policies: optimal power and rate adaptation (opra), optimal rate adaptation with constant power (ora), truncated channel inversion with fixed rate (tifr), and complete channel inversion with fixed rate (cifr). We give numerical examples illustrating the main trends and offer comparisons on the behavior of opra, ora, tifr, and cifr under variation of different parameters. 1. 0 0 0 Bookmarks 22 Views • Source Article: Capacity of Rayleigh fading channels under different adaptive transmission and diversity-combining techniques [hide abstract] IEEE Transactions on Vehicular Technology 08/1999; · 2.06 Impact Factor • Source  Article: Capacity of fading channels with channel side information [hide abstract] ABSTRACT: … |
| **7B (Top 0.10%)** | **7B (Top 97.95%)** |
| ## Sunday, February 8, 2009\n\n### 6. How Euler Derived the Continuity Equation\n\n[Previous Article: The Reynolds Transport Theorem]\n\nI thought that it would be interesting to present Euler's derivation of the continuity equation for incompressible flows. Although d'Alembert, in 1752, had already presented an equivalent equation in his Essai d'une nouvelle théorie de la résistance des fluides (which he had already submitted to the Academy of Sciences of Berlin in 1749), the one proposed by Euler in 1756 (written 1752) is considered to be the most rigorous.\n\nEuler's contribution to Fluid Mechanics goes beyond what a scientist may imagine, and was mostly due to four manuscripts published between 1752 and 1755. These are\n\n1. Principia Motus Fluidorum (1756) [pdf]\n2. Principes généraux de l'état d'équilibre des fluides (1755) [pdf]\n3. Principes généraux du mouvement des fluides (1755) [pdf]\n4. Continuation des recherches sur la théorie de mouvement des fluides (1755) [pdf]\n\nThe final thing I would like to point out is that Euler's genius lies partly in his ability to synthesize and introduce world class notation. In this way, he was able to supersede all his predecessors.\n\nEuler starts by saying:\n\n"… I shall posit that the fluid cannot be compressed into a smaller space, and its continuity cannot be interrupted. I stipulate without qualification that, in the course of the motion within the fluid, no empty space is left by the fluid, but it always maintains continuity in this motion…" [Paragraph 6, Principia Motus Fluidorum, Translated by Enlin Pan]\n\nHe then argues that if one considers any part of a fluid of this type (i.e. incompressible), then each individual particles fill the same amount of space as they move around. He then infers that if this happens for particles, it should happen to the fluid as a whole (which was his assumption of incompressibility). One is now able to consider an arbitrary fluid element and then track its instantaneous changes "to determine …… | Gitlab-runner (docker-machine) concurency and request-concurrency? Can anyone tell me how to set on gitlab-runner ( docker-machine ) parameters: – limit –request-concurrency –machine-idle-nodes concurency (cannot be set from CLI) ? Is --request-concurrency same as concurency parm but just for docker-machine executor ? I would like to have 2 idle nodes, 3 parallel jobs per node and max limit of nodes 10. I am getting WARN message: WARNING: Specified limit (10) larger then current concurrent limit (1). Concurrent limit will not be enlarged. Thanks EDIT: concurency should be number of cores + 1 ? and also concurency=request-concurrency ? |

Figure 17: The cases of AttentionInfluence in OpenWebMath domain.

| 1.3B (Top 0.10%) | 1.3B (Top 97.95% ) |
|---|---|
| import pprint\n\nboard = [\n   [7,8,0,4,0,0,1,2,0],\n   [6,0,0,0,7,5,0,0,9],\n [0,0,0,6,0,1,0,7,8],\n   [0,0,7,0,4,0,2,6,0],\n   [0,0,1,0,5,0,9,3,0],\n [9,0,4,0,6,0,0,0,5],\n   [0,7,0,3,0,0,0,1,2],\n   [1,2,0,0,0,7,4,0,0],\n [0,4,9,2,0,6,0,0,7]\n]\n\ndef solve(brd):\n   """\n   Solves a sudoku board using backtracking\n   :param brd: 2d list of ints\n   :return: solution\n   """\n   find = find_empty(brd)\n   if not find:\n      return True\n   else:\n row, col = find\n\n   for i in range(1,10):\n      if valid(brd, i, (row, col)):\n brd[row][col] = i\n\n         if solve(brd):\n            return True\n\n brd[row][col] = 0\n\n   return False\n\ndef valid(brd, num, pos):\n   # Check row\n   for i in range(len(brd[0])):\n      if brd[pos[0]][i] == num and pos[1] != i:\n         return False\n\n   # Check column\n   for i in range(len(brd)):\n      if brd[i][pos[1]] == num and pos[0] != i:\n return False\n\n   # Check box\n   box_x = pos[1] // 3\n   box_y = pos[0] // 3\n\n   for i in range(box_y*3, box_y*3 + 3):\n      for j in range(box_x * 3, box_x*3 + 3):\n         if brd[i][j] == num and (i,j) != pos:\n            return False\n\n   return True\ndef print_board(brd):\n   for i in range(len(brd)):\n      if i %3 == 0 and i !=0:\n         print("--------------------")\n\n      for j in range(len(brd[0])):\n         if j % 3 == 0 and j != 0:\n print("|", end="")\n\n         if j == 8:\n            print(brd[i][j])\n else:\n            print(str(brd[i][j]) + " " , end="")\n\ndef find_empty(brd):\n for i in range(len(brd)):\n      for j in range(len(brd[0])):\n         if brd[i][j] == 0:\n            return (i, j)\n   \n   return None\n | Bitcoin mining is a process of verifying transactions and recording them on the blockchain ledger. The blockchain is a decentralized public ledger that keeps a record of all Bitcoin transactions. Mining involves solving complex mathematical problems using specialized software and hardware. Explore qumasai.io for further information.

The Bitcoin network rewards miners for successfully verifying transactions by giving them newly created Bitcoins. The mining process involves adding a new block of transactions to the blockchain every 10 minutes. Miners compete against each other to add the next block to the chain.

To participate in Bitcoin mining, one needs to have a powerful hardware setup and specialized mining software. The hardware required is called an ASIC miner, which is specially designed to solve the mathematical problems required to add a block to the blockchain. |

| 7B (Top 0.10%) | 7B (Top 97.95%) |
|---|---|
| #URL: https://leetcode.com/explore/learn/card/hash-table/187/conclusion-hash-table/1134/\n#Description\n"""\nGiven four integer arrays nums1, nums2, nums3, and nums4 all of length n, return the number of \ntuples (i, j, k, l) such that:\n0 <= i, j, k, l < n\nnums1[i] + nums2[j] + nums3[k] + nums4[l] == 0\n\nExample 1:\n\nInput: nums1 = [1,2], nums2 = [-2,-1], nums3 = [-1,2], nums4 = [0,2]\nOutput: 2\nExplanation:\nThe two tuples are:\n1. (0, 0, 0, 1) -> nums1[0] + nums2[0] + nums3[0] + nums4[1] = 1 + (-2) + (-1) + 2 = 0\n2. (1, 1, 0, 0) -> nums1[1] + nums2[1] + nums3[0] + nums4[0] = 2 + (-1) + (-1) + 0 = 0\n\nExample 2:\n\nInput: nums1 = [0], nums2 = [0], nums3 = [0], nums4 = [0]\nOutput: 1\n\nConstraints:\n\nn == nums1.length\nn == nums2.length\nn == nums3.length\nn == nums4.length\n1 <= n <= 200\n-228 <= nums1[i], nums2[i], nums3[i], nums4[i] <= 228\n"""\ndef fillSum(nums1, nums2):\n   sz = len(nums1)\n   sum12 = {}\n   for i in range(sz):\n      for j in range(sz):\n         sm = nums1[i] + nums2[j]\n         if sm in sum12:\n            sum12[sm].append((i, j))\n         else:\n sum12[sm] = [(i, j)]\n   return sum12\ndef fourSumCount(nums1, nums2, nums3, nums4):\n   sum12 = fillSum(nums1, nums2)\n   sum34 = fillSum(nums3, nums4)\n   count = 0\n   for sm in sum12:\n      if -sm in sum34:\n         count += len(sum12[sm]) * len(sum34[-sm])\n   return count | # --*--coding:utf-8#
#Author:cnn\nfrom time import sleep\nimport Multiprocessing

g_num = 0\\

# \nmutex = multiprocessing.Lock()
# \nclass MutiProcess(multiprocessing.Process):
def print_name(self, num):
global g_numfor i in range(0, num + 1):
# mutex.acquire()
g_num += imutex.release()
print(g_num)
sleep(1)
def run(self):
self.print_name(100)
if __name__ == '__main__':
    mu1 = MutiProcess()
    mu2 = MutiProcess()
    mu1.start()
    mu2.start()
    # --*--coding:utf-8 |

Figure 18: The cases of AttentionInfluence in Python-Edu domain.

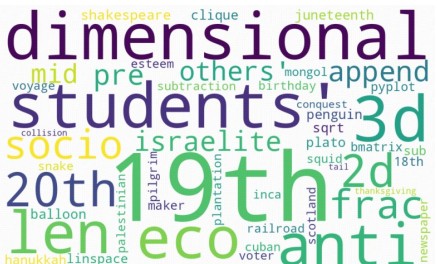 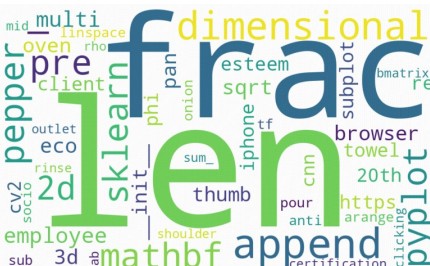

Figure 19: The cloud maps of the data selected by AttentionInfluence and FineWeb-Edu Classifier, respectively. The left part is the cloud map of FineWeb-Edu Classifier, the right part is that of AttentionInfluence.

## O  HIGH FREQUENCY WORDS

| Ranking (%) | Static Method | Data Source | | | |
|---|---|---|---|---|---|
| | | FineWeb-Edu-dedup | Cosmopedia-v2 | Python-Edu | OpenWebMath |
| 10 | TF | 0.84 | 0.73 | 0.29 | 0.57 |
| | TF-IDF | 0.82 | 0.72 | 0.38 | 0.52 |
| 20 | TF | 0.88 | 0.81 | 0.41 | 0.67 |
| | TF-IDF | 0.87 | 0.80 | 0.43 | 0.63 |
| 50 | TF | 0.95 | 0.91 | 0.67 | 0.79 |
| | TF-IDF | 0.92 | 0.90 | 0.66 | 0.78 |

Table 13: Word overlap by ranking threshold and frequency-based statistical method

We separately select the top 10%, 20%, and 50% of samples ranked by AttentionInfluence and the FineWeb-Edu classifier, and compute the overlap of high-frequency words using multiple statistical approaches.

As shown in Table 13, we derive several key insights: 1) AttentionInfluence exhibits a high degree of overlap with the FineWeb-Edu Classifier, highlighting the **reliability of the samples selected by AttentionInfluence**. 2) **AttentionInfluence and the FineWeb-Edu Classifier demonstrate a degree of complementarity**. We observe notable domain-specific variations. Specifically, in the FineWeb-Edu-dedup and Cosmopedia-v2 domains, the overlap exceeds 70%, whereas in the Python-Edu and OpenWebMath domains, it falls below 60%. To further examine the differences between AttentionInfluence and FineWeb-Edu Classifier in specific domains, we sample representative examples from the Python-Edu and OpenWebMath domains, as shown in Appendix L. These cases reveal that although the two methods display different preferences across domains, both yield reasonable selections."

As shown in Table 12 of Appendix O, **AttentionInfluence places greater emphasis on method-related terminology, while FineWeb-Edu Classifier is more sensitive to numerical expressions**. We identify two distinctive high-frequency terms: *"19th"* from subset selected by FineWeb-Edu Classifier and *"sklearn"* from AttentionInfluence's subset. We then retrieve representative documents from the original corpus containing these terms. The sample containing *"19th"* is related to historical topics, whereas the one with *"sklearn"* discusses K-Nearest Neighbors Classifier and Hyperparameter Tuning. **This suggests that AttentionInfluence prefers samples containing hands-on coding or procedural mathematical reasoning.**

As illustrated in Figure 19, we visualize the respective word clouds of AttentionInfluence-1.3B and the FineWeb-Edu Classifier after removing overlapping high-frequency words in the Cosmopeida-V2 domain. The resulting word clouds clearly highlight their distinct focal points, indicating a complementary relationship between the two models. To gain deeper insights, we further examine representative samples corresponding to the key terms in each word cloud.

| Specific Word: sklearn | Specific Word: 19th |
|---|---|
| K-Nearest Neighbors Classifier and Hyperparameter Tuning

In this chapter, we will explore the K-Nearest Neighbors (KNN) classifier, a fundamental machine learning algorithm, and learn how to optimize its performance by tuning hyperparameters. We will use Python, along with popular libraries such as pandas, NumPy, scikit-learn, and matplotlib.

K-Nearest Neighbors Classifier

The KNN classifier is a simple yet powerful algorithm used for both classification and regression tasks. It is a type of instance-based learning, …

First, let's import the necessary libraries:
\begin{verbatim}
 import pandas as pd
import numpy as np
from sklearn.model_selection import train_test_split
from sklearn.neighbors import KNeighborsClassifier
import matplotlib.pyplot as plt
 \end{verbatim}

Next, we will load our dataset, which is a pandas DataFrame df containing the columns 'cases' and 'date'. We will use only these two columns for our analysis: … | Chapter Title: Discovering Sacred Solo Voices in MusicImagine walking into a grand cathedral, dimly lit with tall stained glass windows casting colorful patterns on the cool stone floors. As you take a deep breath, a single voice fills the air, resonating off the walls and ceilings. This soloist sings sacred music – songs written specifically for worship services or religious ceremonies. Through this chapter, we'll embark on an adventure exploring different types of sacred solo voices in various cultures and time periods.Section 1: Gregorian Chant - Monks and Nuns Singing Prayers---In medieval Europe (around 500–1400 AD), monks and nuns created simple yet powerful chants called Gregorian chants. These were sung during Catholic Masses as they believed singing was praying twice! They used only one melody line, which meant everyone sang together in unison. Listen to an example here: <https://www.youtube.com/watch?v=zgYQE7jxx28>. How does it make you feel?Section 2: Indian Classical Music - Exploring Ragas---Let's travel across continents to explore India's rich tradition of classical music. Unlike Western music, Indian classical music focuses heavily on improvisation within specific rules. One popular form is called khayal, where a singer performs a rag (melodic framework) accompanied by a drone instrument like the tanpura. Over centuries, many great singers have developed unique styles passed down generations. Check out this captivating clip featuring renowned vocalist Kishori Amonkar performing a raga based on love.Section 3: Spirituals \& Gospel - From Slaves to Freedom Fighters---During the dark period of slavery in America (16th-19th centuries), enslaved Africans preserved their heritage through secretive gatherings filled with song and dance. Their spirituals often contained … |

Figure 20: The sample of a doc containing the specific word selected by AttentionInfluence-1.3B (**left**) and FineWeb-Edu Classifier (**right**).

# P  LIMITATIONS AND OPPORTUNITIES

While our experimental results demonstrate the effectiveness of AttentionInfluence, several important aspects warrant further investigation. We identify five key areas for future research:

- Our current experiments demonstrate the effectiveness of AttentionInfluence up to 7B parameters and 1,000B tokens of training budget. Extending this approach to long-horizon training and larger-scale models requires a highly expensive computational cost, and we leave it for future research.

- Due to limited manpower, we do not investigate the effects of selected data by AttentionInfluence on the final performance of models, followed by post-training based on open-source data. However, we have conducted supervised fine-tuning (SFT) using our in-house SFT dataset. In this experiment, AttentionInfluence still demonstrated advantages over the baseline—this finding further supports our subsequent hypotheses. Specifically, we hypothesize that reinforcement learning will amplify the good effects of selected data by AttentionInfluence. Furthermore, we believe that AttentionInfluence can be adapted beyond pretraining and extended to the post-training phase, including supervised fine-tuning (SFT) and reinforcement learning (RL), by identifying high-impact training examples that align with model behaviors and target objectives.

- While this work focuses on selecting data from short texts, AttentionInfluence can be readily extended to long texts to identify high-quality samples characterized by long-range dependencies.

- We conduct experiments with alternative approaches for identifying important attention heads, such as the methods proposed by Wu et al. (2024); Fu et al. (2024), which produces a partially overlapping yet distinct set of heads compared to ours. Training LLMs based on the data selected by these heads achieves comparable downstream evaluation performance. More recently, Zhu et al. (2025); Zhang et al. (2025) introduces other compatible methods that can be incorporated into our framework.

  These results demonstrate that AttentionInfluence serves as a flexible and general framework: by defining an appropriate proxy task, one can identify task-relevant attention heads and select associated data via masking. The entire pipeline operates without any supervision signals and is modular by design, allowing the proxy task to be easily replaced depending on the target domain or task. Moreover, the framework is effective even when applied to small pretrained language models, making it practical and scalable for a wide range of data selection scenarios.

  More comprehensive proxy tasks can also be designed to better capture specific types of data within the AttentionInfluence framework, further expanding its applicability and customization potential.

  Furthermore, rather than designing specific proxy tasks, we can perform an exhaustive traversal by systematically disabling each model head across a variety of existing benchmarks. This brute-force approach may allow us to pinpoint key heads and discover the data that drive improvements in model performance.

- The combined effect of multiple heads remains unknown. Moreover, this work does not involve research on the MLP. Substantially more in-depth research endeavors are required to unearth the more fundamental and intrinsic mechanisms underpinning language models.

