# OpenReview forum: "AttentionInfluence: Adopting Attention Head Influence for Weak-to-Strong Pretraining Data Selection"
_ICLR.cc/2026/Conference — Submitted to ICLR 2026_

### Official Review · Reviewer_S3Rg · 2025-10-26

**Soundness:** 3
**Presentation:** 3
**Contribution:** 2
**Rating:** 4
**Confidence:** 4

**Summary:**

In this paper, the authors proposed to use a pretrained LLM to select high-quality and reasoning-intensive pretraining data. Specifically, they first find the retrieval heads of a small model, and then calculate the loss gap with or without these retrieval heads. A higher loss gap indicates a higher reasoning intensity of the data. Experiments have demonstrated the effectiveness.

**Strengths:**

1. The paper has a clear structure and is easy to understand.
2. The proposed method has good practical application scenarios.

**Weaknesses:**

1. The experimental design may not be reasonable enough. Compared to the baseline, the training data is mixed with additional screened 73B data. Should the baseline data also include randomly sampled 73B data?

2. Lack of further experimental analysis. In order to further validate the practical application value of the proposed method, the following analysis may be necessary:

2-1. Is the search head consistent across different corpus data? If not, is it necessary to conduct targeted searches for specific language materials?

2-2. Do the screening model and training model need to be from the same series? For example, can the data filtered by Llama be used to train Qwen?

2-3. In practical applications, CPT data filtering may be a more common scenario. In this scenario, how effective is the proposed method? For example, in CPT training that requires enhanced reasoning ability, the baseline model trained on 400B corpus, while the comparison method trained on high-quality 100B corpus filtered from 400B corpus. If the performance of the comparison method can actually reach or even exceed that of the baseline model, it can demonstrate greater practical value.

2-4. Performance and efficiency analysis of different screening models.

**Questions:**

Please see the weaknesses.

---

> ### Author Response · Authors · 2025-11-16
> **[1/n] Response**
>
> 1. Our baseline in fact already uses 241B data plus an additional 73B tokens of randomly sampled data, and is trained to 1T tokens. Therefore, the comparison is indeed fair.
> 2. **Our paper discusses retrieval heads, not search heads**.
> 3. For the proxy dataset used to detect retrieval heads, if the proxy dataset changes, the detected retrieval heads would naturally change as well. We provide detailed descriptions of our synthetic proxy dataset and its construction process in Section 4.1 (page 5) and Appendix A: Synthetic Test Sample (page 15).
> 4. However, for the target training corpus to be filtered, the set of selected retrieval heads is **kept consistent**. For example, when filtering either the SmolLM corpus or the Nemo-CC [1] corpus, we use the same set of retrieval heads. Once retrieval heads are detected on the proxy dataset, this set becomes fixed and remains consistent throughout, forming a **constant and static data selection model**.
> The SmolLM corpus we use includes four data domains:
>
>
> **FineWeb-Edu-dedup**: English general web pages
>
>
> **Cosmopedia-V2**: LLM-generated encyclopedic data
>
>
> **Python-Edu**: scraped Python code data
>
>
> **OpenWebMath**: scraped mathematical data
>
>
> We apply the same retrieval head set across all four domains when scoring data. Thus, there is **no need to re-detect domain-specific heads** for different domains, genres, or data types—the retrieval heads identified on the proxy dataset are already sufficient. This demonstrates the simplicity, stability, and generality of our method.
>
> 5. The screening model and the training model **do not** need to be from the same series. They can be trained on completely different datasets, and their architectures can also be entirely different—as long as the screening model has **attention heads**, it is sufficient.
>
> In our internal experiments, we conducted even more aggressive tests: **we used a Llama model as the screening model to filter data, and then used the filtered data to train an MoE model. The results were still very strong**.
>
> Regarding the question of whether data filtered by Llama can be used to train Qwen: **Llama and Qwen share highly similar dense Transformer architectures**. Therefore, if Llama is used as the screening model, the filtered data **can absolutely be** used to train Qwen. In fact, our internal experiments already validated a more extreme scenario—using a dense model as the screening model and using the filtered data to train an MoE model—with consistently good results.
>
>
> [1] Su et al. (2024). Nemotron-CC: Transforming Common Crawl into a Refined Long-Horizon Pretraining Dataset.

---

> > ### Author Response · Authors · 2025-11-16
> > **[2/n] Response**
> >
> > 1. We agree with your point that CPT data filtering may be a more common scenario, and our method is indeed well-suited for CPT data filtering.
> >
> >
> > 2. However, we designed our experiments as pretraining rather than CPT data filtering for several considerations:
> >
> >
> > 3. Most open-source models are released as overtrained checkpoints, i.e., trained on an extremely large number of tokens (e.g., Qwen3-0.6B trained on 36T tokens). Such models are not ideal for CPT experiments because the room for improvement is very limited, making it difficult to validate the effect of data selection. Moreover, intermediate checkpoints—which offer more room for improvement—are rarely released in open-source models, and these are more suitable for evaluating data selection effects in CPT settings.
> >
> >
> > 4. CPT is very sensitive to data distribution shifts. Typically, a smooth transition in data distribution is needed, aligned with the original checkpoint’s trained data proportions, to ensure stable and effective CPT training. This is necessary for fair and reliable comparison of CPT data to evaluate data selection. Unfortunately, open-source models usually provide only coarse-grained information about data proportions and do not release the exact checkpoints with their trained data. This makes it difficult to perform CPT experiments under open-source settings.
> >
> >
> > 5. In contrast, pretraining from scratch allows us to precisely control the data proportions. To simulate CPT data filtering, we pretrain from 0–746B tokens (with learning rate warmup 0–10B, then constant 10–746B). We randomly sample 254B tokens from the SmolLM corpus (241B) plus the selected 73B tokens, load the 746B checkpoint, and train on these 254B tokens. This ensures smooth data distribution transitions, addressing the issues above, and is essentially equivalent in principle to CPT data filtering.
> >
> >
> > 6. Therefore, our experimental design is a practical compromise, considering the limitations of available open-source CPT baselines and computational costs. We believe this design demonstrates that our method can be effectively applied to both pretraining data selection and CPT data selection.

---

> > > ### Author Response · Authors · 2025-11-16
> > > **[3/n]  Response**
> > >
> > > 1. In fact, in Section 6.3, “Scalability of AttentionInfluence”, we analyzed the performance of different screening models. **In our main experiments, AttentionInfluence-1.3B refers to using a 1.3B dense model as the screening model to select data, and then training a 7B dense model. We also conducted an ablation experiment, AttentionInfluence-7B, where a 7B dense model is used as the screening model to select data, and then train a 7B dense model.** In Section 6.3, we analyzed the differences in the data filtered by the 1.3B and 7B dense models as screening models. **We found that data selected by the 7B dense model is of higher quality, more diverse, and better generalized.** Comparing the evaluation metrics after training with AttentionInfluence-1.3B and AttentionInfluence-7B, due to space limits, we report detailed results in Appendix I: Detailed Performance Evolution During Pretraining (page 19), Figure 7 (page 19), and Table 8 (page 22). From Figure 7 and Table 8, it is clear that once training surpasses 350B tokens, AttentionInfluence-7B consistently outperforms AttentionInfluence-1.3B on average. **This indicates that as the screening model’s scale increases, the selected data becomes higher quality, more diverse, and better generalized, resulting in improved training performance.**
> > >
> > >
> > >
> > > 2. Regarding efficiency, we also analyze this in the paper. Due to space limits, details are in Appendix G: Experiment Setting, “Computation Cost of AttentionInfluence on SmolLM corpus” (page 17). Specifically:
> > >
> > > Using the 1.3B model, we compute AttentionInfluence scores for all samples in the SmolLM corpus (241B tokens) with 128 A100 GPUs (16 machines, each with 8 A100-80GB GPUs and 900GB CPU memory), taking approximately 5 hours.
> > >
> > >
> > > Using the 7B model, the same computation requires 160 A100 GPUs (20 machines, each with 8 A100-80GB GPUs and 900GB CPU memory) and takes about 25 hours.
> > >
> > >
> > > Clearly, inference with the 7B dense screening model is slower than with the 1.3B model, requiring roughly 5.25× more compute. **However, since data selection is a one-time inference cost, the additional compute is reasonable and justified given the improved training outcomes.**

---

> ### Author Response · Authors · 2025-11-24
>
> Dear reviewer S3Rg,
>
> We appreciate the constructive feedback from you and the time you spent reviewing our work.
>
> We have carefully considered your comments and have endeavored to address these concerns in the response. As the discussion session draws to a close, we would greatly appreciate your guidance on whether our rebuttal adequately resolves your concerns. Additionally, we are more than willing to respond to any inquiries and address any feedback you may have.
>
> Thanks for your time and consideration.

---

### Official Review · Reviewer_7Ecp · 2025-11-01

**Soundness:** 2
**Presentation:** 1
**Contribution:** 2
**Rating:** 2
**Confidence:** 3

**Summary:**

This paper aims to improve data selection for pre-training by leveraging signals from attention heads. Specifically, the authors analyze how large language models allocate attention during reasoning and generation, and introduce a new metric, the AttentionInfluence Score, which quantifies the relative importance of tokens and data based on their attention contributions. The proposed approach first identifies attention heads that are critical for reasoning, masks them in a reference model, and then computes the attention differences between the base and reference models to measure the influence of the data. The authors use the 1.3B model for data selection on the SmolLM corpus, and then pretrain the 7B model on the combined corpus of SmolLM and the selected data instances, showing that the proposed data selection strategy outperforms relevant baselines.

**Strengths:**

* The design of the proposed metric (AttentionInfluence Score) is convincing for data selection.
* The proposed data selection process outperforms relevant baselines.

**Weaknesses:**

* The samples used to identify the important attention heads are very important for the later data selection, as the data instances for pre-training are selected mostly from their signals. In Section 4.1, the authors mention that it is derived from 800 synthetic samples, and it is questionable whether the selected data instances are just very similar to those synthetic samples. Also, more details on constructing those samples and their quality should be provided. Lastly, it would be great if the authors could justify why only the top 5% of the attention heads are selected for data selection.
* It is not intuitive that the authors select the pre-training data from the SmolLM corpus and then use the SmolLM corpus + the selected data instances from the same SmolLM corpus. In other words, the selected data instances for pre-training are just a subset of the SmolLM corpus (which is also used for pre-training), and it seems this setting just unsamples the existing data rather than demonstrating true data selection benefits.
* For pre-training research, it would be great to show the scaling law as a function of the number of parameters, in addition to the number of tokens provided.
* It is unclear why the authors report the results without learning rate decay settings in the main tables.
* Recent training strategies of modern foundation models include mid- and post-training. It is questionable whether the pre-trained model with the proposed data selection strategy can still be effective after mid- and post-training. In addition to this, it would also be interesting to see whether the proposed data selection strategy can be beneficial for the mid- and post-training stages, where the data selection process is typically more rigorous than the pre-training stage.
* The term Llama2-like-1.3B model is unclear. Is this not the Llama2 model?
* In Line 204, two references are broken.

**Questions:**

Please see Weaknesses above.

---

> ### Author Response · Authors · 2025-11-15
> **[1/n] Here we provide a synthesized illustrative example.**
>
> Here we provide a synthesized illustrative example. To comply with the maximum character limit, we have replaced multiple key-value pairs in the sample with '......'. The sample is shown below:
> ```
> Please extract the values corresponding to the specified keys in the JSON object below. Only output the value for the given key, and output nothing else.
>
> The JSON data is as follows: {"lXD2Kqpr7os0Fup1CuSPGEkxox2pDG2X": "If you were truly a fairy I still wouldn’t be able to handle it. I’m not expecting you to look like a magazine cover model whose face stuns people at first sight. A normal person who looks fashionable on the outside, conservative inside, and healthy in both body and mind is enough. If she’s also a bit gentle, that’s even more reliable. I like women who can fold clothes — after washing and ironing, they fold them so neatly it's like they just came from the store.", "fwgFNTqlhSES1SPh3htZ6CMRRv28OlOG": "In a quantum system in quantum mechanics, what physicists are most interested in is finding the ground state of the system, which is the state with the smallest energy eigenvalue. For example: in a quantum system formed by two spin-1/2 particles, if the interaction between the particles can be written as formula_1 where formula_2, formula_3, formula_4 denote the Pauli matrices of the formula_5-th spin. Diagonalizing the above 4×4 matrix yields the eigenvalues: formula_6, and the corresponding eigenvectors are formula_", ......, "GIVABvx0EbzOdramaVgnRLgdy0M5NqbC": "Your older sister’s son does indeed call you jiùjiu (maternal uncle), but your older brother’s son calls you shūshu (paternal uncle), so he’s still called a wàisheng (nephew on the mother’s side).", "7I0e2ud8b2rtb6YoJODELDlzvcPqAhTO": "Based on the problem statement, walking one lap around the square piece of land is 1200 / 3 = 400 1200/3=400 meters, so the side length of the square is 400 / 4 = 100 400/4=100 meters. Therefore, the area of the square land is 100 × 100 = 10,000 100×100=10,000 square meters. Because 1 hectare equals 10,000 square meters, the area of the square land is 1 hectare."}
>
> You need to answer the following questions:
>
> Question: What is the element corresponding to the key 7I0e2ud8b2rtb6YoJODELDlzvcPqAhTO?
> Answer: Based on the problem statement, walking one lap around the square piece of land is 1200 / 3 = 400 1200/3=400 meters, so the side length of the square is 400 / 4 = 100 400/4=100 meters. Therefore, the area of the square land is 100 × 100 = 10,000 100×100=10,000 square meters. Because 1 hectare equals 10,000 square meters, the area of the square land is 1 hectare.
>
> Question: What is the element corresponding to the key Y6Mis8l223EZspkjebYWuNy4mvFFJ9Wd?
> Answer: In a quantum system in quantum mechanics, what physicists are most interested in is finding the ground state of the system, which is the state with the smallest energy eigenvalue. For example: in a quantum system formed by two spin-1/2 particles, if the interaction between the particles can be written as formula_1 where formula_2, formula_3, formula_4 denote the Pauli matrices of the formula_5-th spin. Diagonalizing the above 4×4 matrix yields the eigenvalues: formula_6, and the corresponding eigenvectors are formula_
>
> Question: What is the element corresponding to the key fwgFNTqlhSES1SPh3htZ6CMRRv28OlOG?
> Answer: Your older sister’s son does indeed call you jiùjiu (maternal uncle), but your older brother’s son calls you shūshu (paternal uncle), so he’s still called a wàisheng (nephew on the mother’s side).
>
> Question: What is the element corresponding to the key lXD2Kqpr7os0Fup1CuSPGEkxox2pDG2X?
> Answer:
> ```

---

> > ### Author Response · Authors · 2025-11-15
> > **[2/n] Response**
> >
> > 1. In the [1/n] response, we provided an example of a synthesized sample, which contains multiple key-value pairs and few-shot QA. Here, the keys are all random hash strings, and the values are text segments randomly sampled from documents on the Internet, resulting in high diversity. Because there are multiple random hash strings and the text segments are sampled independently, their contents are unrelated. Therefore, the 800 synthetic samples are **very different** from samples in the SmolLM corpus, where the samples are natural text and do not contain multiple random hash strings or concatenations of unrelated text fragments.
> >
> > 2. In Section 4.1, we provide the design idea for synthesizing these samples. Due to space constraints, we placed the template for the synthesized samples in Appendix A, SYNTHETIC TEST SAMPLE, on page 15 of the paper. Here, we describe the construction process in detail:
> > Randomly generate a hash string as the key, then sample a document from the corpus (without replacement) as the value. Repeat this process multiple times until the total number of tokens is close to 3072, so that these multiple key-value pairs together form the final JSON document.
> > Randomly select 4 keys (without replacement). Since we have recorded the value corresponding to each key, we format the 4 keys as follows:
> > ```
> > Question: What is the element corresponding to the key {key1}?
> > Answer: {value1}
> >
> > Question: What is the element corresponding to the key {key2}?
> > Answer: {value2}
> >
> > Question: What is the element corresponding to the key {key3}?
> > Answer: {value3}
> >
> > Question: What is the element corresponding to the key {key4}?
> > Answer:
> > ```
> >
> > Since this synthesized sample is created based on rules and does not rely on any model, it is highly reliable. Moreover, the correctness of the QA pairs can be guaranteed 100%, so these synthesized samples can be considered high-quality samples suitable for testing a model's retrieval heads.
> >
> > Moreover, due to space constraints, in Appendix F, ABLATION STUDIES ON THE IDENTIFICATION OF IMPORTANT HEADS (page 16), we present the effects of using different datasets on the identification of important (here we refer to retrieval) heads. It can be observed that, compared with the original paper on retrieval heads [1], the retrieval heads identified using our synthesized samples overlap 70.6% with those identified using the synthetic data provided in that paper, demonstrating the reliability of our synthesized data. The reason we designed this method is that pretrained models may only support a context length of 4096, and our approach allows precise control over the length of the synthesized data.
> >
> >
> > 3. Regarding why we select only the top 5% of attention heads for data selection, our paper states:
> > Referring to the original paper on retrieval heads, we select the heads ranked in the top 5% as specifically important heads.
> > We follow the original retrieval-head paper, which shows that masking the top 5% of retrieval heads causes a severe drop in retrieval and reasoning performance. We therefore adopt the same setting for two reasons: (1) it keeps our method simple without introducing an additional hyperparameter to tune, and (2) selecting the top 5% is empirically effective in our experiments.
> >
> > [1] Wu et al. (2024). Retrieval Head Mechanistically Explains Long-Context Factuality.

---

> > > ### Author Response · Authors · 2025-11-15
> > > **[3/n] Why we use the SmolLM corpus + the selected data instead of the selected data**
> > >
> > > The reason we use the SmolLM corpus + the selected data is as follows. To control both the inference cost of data selection and the training cost of the pretraining experiments, we choose the SmolLM corpus, which is a moderately sized dataset containing 241B tokens. We then select the top 20% of samples at the sample level, which corresponds to 73B tokens. However, 73B tokens alone are far from sufficient to train a full 7B model (i.e., in scaling-law terms, 𝐷≪𝑁).
> > >
> > > If the SmolLM corpus had 2410B tokens, then the selected subset (730B tokens) would be large enough to support using only the selected data—rather than using the SmolLM corpus + the selected data. But clearly, with such a large corpus, both the inference cost of data selection and the training cost of the pretraining experiments would be prohibitively high for academic research.
> > >
> > > Based on the considerations above, for our method and all other baselines in the paper (except the most basic baseline), we consistently use SmolLM corpus + the selected data as the training data. This unified setting ensures fairness.
> > >
> > > For the most basic baseline, we use the full 241B-token corpus and continue training until 1T tokens—that is, without upsampling any biased subset. The experimental results show that our method, as well as all other baselines, outperform this most basic baseline. This demonstrates that upsampling a certain good subset of data (typically higher-quality data) brings benefits.
> > >
> > > Our method helps identify higher-quality data from the original corpus, and models trained on this data achieve higher evaluation metrics than the most basic baseline. This indicates that the data identified by our method is of higher quality than random sampling, thereby supporting the effectiveness and benefits of our approach as a data selection method.

---

> > > > ### Author Response · Authors · 2025-11-15
> > > > **[4/n] Response**
> > > >
> > > > I agree with your point. However, due to the limited computational resources available for this paper, we had two possible options:
> > > > 1. Train 340M / 680M / 1.3B models each for 1T tokens
> > > > 2. Train a 7B model for 1T tokens
> > > >
> > > > and we could only choose one. Considering that the goal of our paper is to demonstrate that high-quality data selected using a small pretrained model (1.3B) can improve the training of larger models—as stated in the title, Weak-to-Strong Pretraining Data Selection—and our smallest available pretrained model is 1.3B, our experiments can only be conducted on models larger than 1.3B.
> > > >
> > > > If we had chosen option 1, although there would be variation in model size, it would not suffice to demonstrate Weak-to-Strong Pretraining Data Selection. Therefore, we chose option 2. Training a model larger than 7B could be possible (e.g., a 13B model), but the computational cost would be prohibitive for academic research.
> > > >
> > > > We believe that conducting experiments at the 7B scale is already sufficient to convincingly demonstrate that high-quality data selected using a small pretrained model (1.3B) can improve the training of larger models. Furthermore, training models smaller than 1.3B could risk introducing a distillation effect; in fact, one could speculate that the benefits of data selection would be even more pronounced in smaller models, but that is not the focus of our study.
> > > >
> > > > In some papers on LLM pretraining, such as In-context Pretraining [1], the experiments were conducted at the 7B scale without experiments on smaller models. This also indirectly suggests that, for pretraining experiments in academic research, 7B is a very credible setting, and in general, results at this scale are more reliable than those obtained from smaller models.
> > > >
> > > > [1] Shi et al. (2023). In-context Pretraining: Language Modeling Beyond Document Boundaries.

---

> > > > > ### Author Response · Authors · 2025-11-15
> > > > > **[5/n] Response**
> > > > >
> > > > > In Table 1, we report the main results on various benchmarks at the middle stage of training (~500B tokens), without applying learning rate decay (LRD). In Table 2, we report the main results after full training (1T tokens), with LRD applied.
> > > > >
> > > > > We adopt a WSD (Warmup-Stable-Decay) learning rate schedule: warmup from 0–10B tokens, stable from 10–746B, and cosine decay from 746–1000B tokens. Training the 7B model to 1T tokens consumed over 20,000 H100 GPU hours (256 GPUs × 80 hours), roughly 70,000 USD on major cloud providers (assuming 3.50 USD per H100 GPU hour). Training all models (Baseline, PPL filter, Scaling Filter, FineWeb-Edu Classifier, AttentionInfluence-1.3B, AttentionInfluence-7B) to 1T would cost ~6 × 70,000 USD.
> > > > >
> > > > > To save resources, we evaluated all experiments at 495B tokens. PPL filter and Scaling Filter were the two worst-performing experiments, significantly behind others. We therefore halted their training, assuming they would remain inferior after full training. This is noted in a footnote on page 6: "Due to limited computational resources, the training of other unsupervised baselines was halted at the middle of pretraining, as they had already fallen behind the strong supervised FineWeb-Edu Classifier."
> > > > >
> > > > > In summary, this explains why we report the results without learning rate decay in the main tables—namely, to observe the evaluation metrics of each experiment early and halt the underperforming ones to save computational resources. We also report the final training results after 1T tokens with learning rate decay applied in the main tables, which fully demonstrates the effectiveness of our method compared to the baselines.

---

> ### Author Response · Authors · 2025-11-15
> **[6/n] Response**
>
> 1. Whether the pre-trained model with the proposed data selection strategy remains effective after mid- and post-training
>
> We consider mid-training as the phase where high-quality or synthetic data, or non-CoT/CoT SFT-like data, is introduced with learning rate annealing. Since no widely accepted open-source dataset of this kind exists, we did not perform a full mid-training. Instead, we applied cosine learning rate decay from 746B–1000B tokens using in-distribution data, which can be regarded as a weakened form of mid-training. At 1T tokens, our method achieves higher evaluation metrics than other baselines.
>
> For post-training, at the time of our experiments (~March–May 2025), there was no publicly available, sufficiently general SFT dataset. A proper SFT dataset would require a complex mix of data types—instruction following, chat, math, code, science, safety, etc.—and single-type datasets such as LIMA [1] or s1 [2] are insufficient for a comprehensive and fair comparison. While NVIDIA’s Llama-Nemotron Post-Training Dataset [3], which was recently released, might be suitable, using it would incur substantial additional experimental cost. Moreover, it would still require a non-trivial investigation into how to set the proportion and number of samples for each domain to make it most suitable for checkpoints pre-trained on the SmolLM Corpus.
>
> During our experiments, we used an in-house SFT dataset, initializing from the baseline 1T checkpoint and our AttentionInfluence-1.3B 1T checkpoint. The evaluation results are as follows:
>
> | Dataset / Model       | Baseline SFT | AttentionInfluence-1.3B SFT |
> |---------------|--------------|------------------------------|
> | MMLU-Pro      | 26.22        | 29.43                        |
> | BBH           | 42.32        | 44.40                        |
> | MATH          | 24.04        | 26.40                        |
> | Avg           | 30.86        | 33.41                        |
>
>
> 2. Whether the proposed data selection strategy can be beneficial for the mid- and post-training stages
>
> We have also explored this question, though it is beyond the scope of this paper. Our paper is titled AttentionInfluence: Adopting Attention Head Influence for Weak-to-Strong Pretraining Data Selection. The design of our method in this paper is intended primarily for pretraining data selection, rather than for post-training data selection.
>
> Nevertheless, we conducted preliminary experiments applying AttentionInfluence to post-training as a preview. Specifically, we applied the method to the s1 dataset, selecting the top 1K high-quality samples to compare with s1K-1.1 (1,000 manually filtered samples), and performed SFT on Qwen2.5-32B-Instruct. The results are as follows:
>
> | Dataset / Model             | s1K-1.1 | AttentionInfluence 1K |
> |-----------------------------|--------|----------------------|
> | AIME 2024                   | 0.5666 | 0.700                |
> | MATH 500                    | 0.944  | 0.946                |
> | GPQA Diamond                | 0.6212 | 0.666                |
> | Avg                         | 0.7106 | 0.7707               |
>
>
> These results provide a preliminary indication that our method can select higher-quality samples for post-training, though this line of investigation can be considered a separate direction for future work.
>
> [1] Zhou et al. (2024). LIMA: Less Is More for Alignment.
>
> [2] Muennighoff et al. (2025). S1: Simple Test-Time Scaling.
>
> [3] Akhiad et al. (2025). Llama-Nemotron: Efficient Reasoning Models (huggingface.co/datasets/nvidia/Llama-Nemotron-Post-Training-Dataset).

---

> > ### Author Response · Authors · 2025-11-15
> > **[7/n] Response**
> >
> > 1. Respone to "The term Llama2-like-1.3B model is unclear. Is this not the Llama2 model?"
> > The Llama2-like-1.3B model refers to an in-house pretrained model adopting the LLaMA 2 architecture. The smallest scale of the open-source LLaMA 2 models is 7B, which is not convenient for efficient inference in academic research. We have an in-house 1.3B model that uses the LLaMA 2 architecture; to achieve a 1.3B parameter size, we adjusted the hidden size, vocabulary size, number of layers, etc., while also adopting a more modern GQA design, which in the official LLaMA 2 series is only used in LLaMA 2-70B. Therefore, we use the term Llama2-like-1.3B to describe this in-house 1.3B model. Due to space constraints, we have provided all the details of the Llama2-like-1.3B model in Appendix G, Experiment Setting, Table 5.
> >
> > 2. Respone to "In Line 204, two references are broken."
> > Sorry, it is a typo and we will revise it in the final version.
> > lin2024rho refers to: [1] Lin et al. (2024). Rho-1: Not All Tokens Are What You Need.
> > ko2024mirrored refers to: [2] Ko et al. (2024) The Mirrored Influence Hypothesis: Efficient Data Influence Estimation by Harnessing Forward Passes.

---

> ### Author Response · Authors · 2025-11-24
>
> Dear reviewer 7Ecp,
>
> We appreciate the constructive feedback from you and the time you spent reviewing our work.
>
> We have carefully considered your comments and have endeavored to address these concerns in the response. As the discussion session draws to a close, we would greatly appreciate your guidance on whether our rebuttal adequately resolves your concerns. Additionally, we are more than willing to respond to any inquiries and address any feedback you may have.
>
> Thanks for your time and consideration.

---

### Official Review · Reviewer_wvBs · 2025-11-01

**Soundness:** 3
**Presentation:** 3
**Contribution:** 3
**Rating:** 6
**Confidence:** 3

**Summary:**

This paper proposes AttentionInfluence, a training-free and unsupervised method for pretraining data selection. The key idea is that data activating more retrieval heads are high-quality and encode reasoning-related behaviors. Using a 1.3B model to select the top ~20% (73B tokens) from the SmolLM corpus (241B tokens) based on the AttentionInfluence score, and mixing them to train a 7B model with 1T total training tokens, the approach outperforms both unsupervised and supervised baselines on reasoning and knowledge benchmarks.

**Strengths:**

1. The paper introduces a new perspective by leveraging mechanistic interpretability (retrieval head behavior) for pretraining data selection.
2. It provides detailed ablations and qualitative analyses.
3. The method is effective as demonstrated by the pretraining experiments while being entirely training-free and unsupervised.

**Weaknesses:**

Since only one pretraining corpus (SmolLM) and one pretraining model (a 7B model) are used, the robustness and generalizability of the method may be limited. Considering the high cost of pretraining and the theoretical generality of the AttentionInfluence method, it should be possible to further verify its effectiveness through post-training experiments.

**Questions:**

1. The full SmolLM corpus contains 241B tokens, and the selected subset adds another 73B tokens, while the total training uses 1T tokens. How many of these 1T tokens come from the selected subset (for both AttentionInfluence and FineWeb-Edu Classifier), and how does this proportion differ from the baseline?
2. Is AttentionInfluence applicable to the mid- or post-training stage? Could you provide results using a smaller and different corpus and a different model at the mid- or post-training stage to verify the robustness of this method?

---

> ### Author Response · Authors · 2025-11-17
> **[1/n] Response**
>
> 1. Whether the pre-trained model with the proposed data selection strategy remains effective after post-training
>
> For post-training, at the time of our experiments (~March–May 2025), there was no publicly available, sufficiently general SFT dataset. A proper SFT dataset would require a complex mix of data types—instruction following, chat, math, code, science, safety, etc.—and single-type datasets such as LIMA [1] or s1 [2] are insufficient for a comprehensive and fair comparison. While NVIDIA’s Llama-Nemotron Post-Training Dataset [3], which was recently released, might be suitable, using it would incur substantial additional experimental cost. Moreover, it would still require a non-trivial investigation into how to set the proportion and number of samples for each domain to make it most suitable for checkpoints pre-trained on the SmolLM Corpus.
>
> During our experiments, we used an in-house SFT dataset, initializing from the baseline 1T checkpoint and our AttentionInfluence-1.3B 1T checkpoint. The evaluation results are as follows:
>
> | Dataset / Model       | Baseline SFT | AttentionInfluence-1.3B SFT |
> |---------------|--------------|------------------------------|
> | MMLU-Pro      | 26.22        | 29.43                        |
> | BBH           | 42.32        | 44.40                        |
> | MATH          | 24.04        | 26.40                        |
> | Avg           | 30.86        | 33.41                        |
>
>
> 2. Whether the proposed data selection strategy can be applied to post-training data selection
>
> We have also explored this question, though it is beyond the scope of this paper. Our paper is titled AttentionInfluence: Adopting Attention Head Influence for Weak-to-Strong Pretraining Data Selection. The design of our method in this paper is intended primarily for pretraining data selection, rather than for post-training data selection.
>
> Nevertheless, we conducted preliminary experiments applying AttentionInfluence to post-training as a preview. Specifically, we applied the method to the s1 dataset, selecting the top 1K high-quality samples to compare with s1K-1.1 (1,000 manually filtered samples), and performed SFT on Qwen2.5-32B-Instruct. The results are as follows:
>
> | Dataset / Model             | s1K-1.1 | AttentionInfluence 1K |
> |-----------------------------|--------|----------------------|
> | AIME 2024                   | 0.5666 | 0.700                |
> | MATH 500                    | 0.944  | 0.946                |
> | GPQA Diamond                | 0.6212 | 0.666                |
> | Avg                         | 0.7106 | 0.7707               |
>
>
> These results provide a preliminary indication that our method can select higher-quality samples for post-training, though this line of investigation can be considered a separate direction for future work.
>
> [1] Zhou et al. (2024). LIMA: Less Is More for Alignment.
>
> [2] Muennighoff et al. (2025). S1: Simple Test-Time Scaling.
>
> [3] Akhiad et al. (2025). Llama-Nemotron: Efficient Reasoning Models (huggingface.co/datasets/nvidia/Llama-Nemotron-Post-Training-Dataset).

---

> > ### Author Response · Authors · 2025-11-17
> > **[2/n] Response**
> >
> > 1. For Q1: For the FineWeb-Edu Classifier, we precisely controlled the top-quality subset it selects and ensured that the total token count of the subset is 73B. For the baseline, we randomly sampled 73B tokens as the subset. Therefore, out of the total 1T tokens, approximately 1000 / (241+73) * 73 = ~232B tokens come from each selected subset. This means that each selected subset is trained ~3.2 times. **In terms of proportion, this is exactly the same as the baseline, ensuring a fair comparison across methods.**
> > 2. For Q2: We consider mid-training as the phase where high-quality or synthetic data, or non-CoT/CoT SFT-like data, is introduced with learning rate annealing. Since no widely accepted open-source dataset of this type exists, we did not conduct a full mid-training. Instead, we applied a cosine learning rate decay from 746B to 1000B tokens using in-distribution data, which can be regarded as a weakened form of mid-training. At 1T tokens, our method achieves higher evaluation metrics than other baselines.
> > Thus, for data selection on high-quality or synthetic mid-training data, our current experiments already indicate applicability. For non-CoT/CoT SFT-like mid-training data, this scenario is more akin to post-training data: as long as the method is effective in the post-training stage, it is highly likely to work here as well. Accordingly, we supplement with post-training experiments, which have been reported in [1/n] response. **In these experiments, we used the s1 dataset and initialized SFT from Qwen2.5-32B-Instruct. This clearly satisfies the condition of using a smaller and different corpus and a different model at the mid- or post-training stage, and the results show that our method is also effective in the post-training stage, with even larger gains.**

---

> ### Author Response · Authors · 2025-11-24
>
> Dear reviewer wvBs,
>
> We appreciate the constructive feedback from you and the time you spent reviewing our work.
>
> We have carefully considered your comments and have endeavored to address these concerns in the response. As the discussion session draws to a close, we would greatly appreciate your guidance on whether our rebuttal adequately resolves your concerns. Additionally, we are more than willing to respond to any inquiries and address any feedback you may have.
>
> Thanks for your time and consideration.

---

### Official Review · Reviewer_3xfu · 2025-11-09

**Soundness:** 3
**Presentation:** 3
**Contribution:** 2
**Rating:** 4
**Confidence:** 4

**Summary:**

The paper proposes AttentionInfluence, a new method for efficient pre-training data selection by leveraging the retrieval heads. AttentionInfluence identifies the important attention heads in a small LLM for retrievals and selects pre-training data examples based on the loss difference over examples between keeping and masking out such attention heads. Experiments show that AttentionInfluence selects data that improves downstream performance on knowledge-intensive and reasoning-intensive tasks, and is more efficient than other data selection baselines, as a small LLM is employed as the data selector.

**Strengths:**

1. The paper proposes a new pre-training data selection method with a focus on the efficiency of data selection and weak-to-strong generalization. Such new perspectives on pre-training data selection contribute to the literature beyond language modeling.

2. The proposed method is well grounded in the interpretability literature, and experiments across multiple benchmarks provide empirical support.

3. The paper presents comprehensive analyses of different design choices associated with the proposed method.

**Weaknesses:**

1. There exists a mismatch between the functionality of retrieval heads (long-context retrieval and reasoning) and the downstream task of the paper (pre-training data selection), and this leads to my concern about whether the proposed method is appropriate and well-motivated. In the literature, the retrieval heads are shown to be important for long-context retrieval, understanding, and reasoning tasks (e.g., needle-in-the-haystack), but their influences on short-context tasks are much less strong. In the pre-training literature, retrieval heads are also discussed more in the context of long-context pre-training or context extension. However, this paper does not specifically target long-context pre-training, and all the downstream tasks being evaluated (e.g., those in Table 1 and Table 2) are short-context tasks. Therefore, in my opinion, there is a mismatch between the methodology and the downstream task in this paper. While the author might have been aware of the effect of context length, as Section 6 shows that AttentionInfluence selects longer data examples, the discussion is rather limited; this paper needs to be better motivated by including more discussions/experiments on the effects of context length.

2. The empirical result is relatively weak compared to the baselines. For example, in Table 1, AttentionInfluence-1.3B is worse on average compared with the FineWeb-Edu Classifier baseline, and < 1% better than the simple PPL filter baseline. In a sense, this is intuitive because of the mismatch in W1: most of the evaluation tasks are short-context, and data selected by leveraging retrieval heads might not show large enough effects for such tasks. I would expect AttentionInfluence to outperform other baselines more on long-context tasks.

3. The analyses depend heavily on loosely defined metrics. Several analyses in Section 6 use the metrics of Education Score and Reasoning score to emphasize the strength of the proposed method. While I appreciate the in-depth analyses present, the two metrics are loosely defined: (1) They are not commonly used metrics in the literature, as I did not find references provided in this paper or relevant papers in the literature that use these metrics, especially the education score; (2) They are not well-defined in the LLM-as-a-judge prompt. As the prompt in Appendix J, there is no definition for the term "educational value" and the definition for the term "reasoning-intensive" is also slightly vague. Given the vague definitions, it is unclear if the LLM-as-a-judge scores accurately capture the desired features of selected data. For example, the education scores in Table 10 and Table 11 clearly saturate and cannot differentiate between different methods at all.

**Questions:**

Please refer to the weakness part.

---

> ### Author Response · Authors · 2025-11-17
> **[1/n] Response**
>
> We thank the reviewer for raising this important point regarding the relationship between retrieval heads, long-context behavior, and our downstream objective of pre-training data selection. Below we provide a detailed clarification and motivation.
>
> 1. Retrieval heads are crucial for chain-of-thought reasoning.
> In the original Retrieval Head paper [1], Section 4.3 “Chain-of-Thought Reasoning Also Requires Retrieval Heads” (page 7) shows that masking the retrieval heads causes the model’s CoT performance to collapse, making the model “blind” to key input information and prone to hallucination. This demonstrates that retrieval heads are indispensable for CoT reasoning. Our work is motivated precisely by this property: we leverage retrieval heads to identify pre-training examples with high reasoning intensity, thereby improving the model’s downstream chain-of-thought ability.
>
> 2. Retrieval heads also explain long-context factuality but this does not contradict our motivation.
> In Section 4.1 of [1], the authors show that retrieval heads are essential to the model’s Needle-in-a-Haystack (NIAH) performance. We agree that NIAH primarily reflects long-context retrieval rather than reasoning.
>
> 3. Reasoning and long-context retrieval are two sides of the same coin.
> Combining the observations above, [1] concludes that retrieval heads are simultaneously crucial for long-context behavior and for reasoning. We view these two abilities as two sides of the same coin: both stem from the same atomic capability of the model—retrieval heads.
> Prior work has applied retrieval heads to reduce KV-cache cost during long-context inference (the idea proposed in [1] and implemented with improvements in [2]). Many follow-up studies explore inference-efficiency benefits, whereas our work opens a new application scenario—pre-training data selection—which, to our knowledge, has not been explored in any retrieval-head-related literature, and this constitutes the primary innovation of our paper.
> We also note that very recent work [3] uses retrieval heads to improve long-context reasoning and re-ranking. Although published later and entirely orthogonal to our approach, it further corroborates the viewpoint that long-context retrieval and reasoning rely on the same mechanism.
>
> 4. Empirical motivation: masking retrieval heads severely harms reasoning tasks.
> Consistent with [1], we also replicated the effect that masking attention heads causes substantial degradation on GSM8K. Due to space constraints, we included our results in Appendix E “Mirror Effects in AttentionInfluence” (page 16). Table 3 shows that masking attention heads significantly degrades performance on reasoning tasks such as GSM8K, BBH, and DROP. Note that DROP can be viewed as a relatively long-context task under our setting: we evaluate it using a 3-shot configuration with a model context length of 8192, and the average length of the evaluation samples already exceeds 1,024 tokens.
> This again confirms that retrieval heads are essential for CoT. Our method is therefore motivated by using retrieval heads as a signal to identify data that significantly improves CoT—and our experiments validate this effect.
>
> [1] Wu et al. (2024). Retrieval Head Mechanistically Explains Long-Context Factuality.
>
> [2] Fu et al. (2024). Not All Heads Matter: A Head-Level KV Cache Compression Method with Integrated Retrieval and Reasoning.
>
> [3] Zhang et al. (2025). Query-Focused Retrieval Heads Improve Long-Context Reasoning and Re-ranking.

---

> ### Author Response · Authors · 2025-11-17
> **[2/n] Response**
>
> **Retrieval heads can also be used for long-context data selection.**
>
> We agree with the reviewer that retrieval heads can, in principle, be used for long-context data selection. Due to space limitations, Appendix P “Limitations and Opportunities” (page 33) states:
> “While this work focuses on selecting data from short texts, AttentionInfluence can be readily extended to long texts to identify high-quality samples characterized by long-range dependencies.”
> We have internally verified that applying AttentionInfluence to long-context data selection improves long-context performance as well. However, this represents another full research direction. For this submission, we prioritized our limited resources toward short-text experiments, since our **off-the-shelf pretrained 1.3B model**, which we adapt into a data-selection model via AttentionInfluence, is trained with only a 4096-token context window, while the 7B target model uses 8192-token sequences. In contrast, a model suitable for long-context data selection would itself require a much larger window size (e.g., 32K), and long-context continual training typically involves very long sequence lengths (e.g., ≥64K) [1]. Both aspects incur substantially higher inference and training compute, so we plan to revisit this with a dedicated long-context experimental setting in future work and release a follow-up paper.
>
> Although this paper reports only short-context evaluation benchmarks, DROP already serves as a relatively longer-context test. In our setup, DROP is evaluated in a 3-shot configuration with a model context window of 8192 tokens, and the average length of its evaluation samples exceeds 1,024 tokens. On DROP, AttentionInfluence-1.3B@1T = 36.52 > Baseline@1T = 36.09 > FineWeb-Edu Classifier = 34.68. Section 6.1 further shows that AttentionInfluence selects data with longer average lengths than the FineWeb-Edu classifier, consistent with the DROP results.
>
> We also evaluated long-context behavior internally using a 32K(32768)-length NIAH extrapolation setting:
> AttentionInfluence-1.3B@1T = 34.1 > FineWeb-Edu = 31.1 > Baseline@1T = 28.0, indicating that our data selection method also improves long-context capability—a point consistent with the reviewer’s intuition.
>
> However, since our pre-training context length is 8192 and long-context evaluation benchmarks typically evaluate models at sequence lengths around 131,072 tokens, which corresponds to the long-document regime targeted in recent studies, we felt that reporting such metrics would be less meaningful within the current experimental setting.
>
> Lastly, our data-selection model is a 1.3B dense model trained with a maximum context of 4096, and the loss is computed over the first 4096 tokens only. Meanwhile, the 7B model is trained with 8192-token sequences. This indicates that AttentionInfluence provides some degree of length extrapolation in data selection.
>
> [1] Gao et al. (2024). How to Train Long-Context Language Models (Effectively).

---

> > ### Author Response · Authors · 2025-11-17
> > **[3/n] Response**
> >
> > 1. As described in Section 5.1 Experimental Details (page 6), the FineWeb-Edu Classifier is a strong supervised and training-required baseline, distilled from LLaMA2-70B-Instruct responses. In contrast, AttentionInfluence-1.3B is a fully unsupervised and training-free data selection model, and it is based only on a dense 1.3B model. The fact that AttentionInfluence-1.3B is already comparable to the FineWeb-Edu Classifier strongly indicates that our method—despite requiring no supervision and no training—is sufficiently effective.
> > Over the full training trajectory (omitted from the main paper due to space limits), the complete evaluation curves are shown in Appendix I: Detailed Performance Evolution During Pretraining (page 19). In Figure 7, AttentionInfluence-1.3B is comparable to the FineWeb-Edu Classifier for most of training (e.g., around 495B tokens), and becomes consistently better during the learning-rate decay phase, ending with a clear advantage at 1T tokens.
> >
> > 2. We report the average score over 21 evaluation datasets as the avg metric. Even though AttentionInfluence-1.3B appears to be only 0.9 points better than the simple PPL filter baseline at 495B tokens, this is already a highly reliable and credible improvement. For reference, the well-known Scaling Filter baseline improves over the simple PPL filter baseline by only 0.27.
> > Your intuition is correct: AttentionInfluence is expected to outperform other baselines more significantly on long-context tasks. As shown in the internal long-context evaluation we reported in our [2/n] response, using a 32K-length NIAH extrapolation setting, we observe:
> >
> >
> > AttentionInfluence-1.3B@1T = 34.1
> >
> >
> > FineWeb-Edu = 31.1
> >
> >
> > Baseline@1T = 28.0
> >
> >
> > The gains in the long-context setting are substantially larger than the average gains over the 21 short-context benchmarks. This is understandable: current long-context evaluations primarily measure retrieval-style behavior, which is more of a vertical capability and relatively easier to improve. In contrast, improving the average performance across 21 diverse short-context benchmarks represents a more challenging—and more credible—achievement.

---

> > > ### Author Response · Authors · 2025-11-17
> > > **[4/n] Response**
> > >
> > > 1. We use the Education Score to quantify the educational value contained in a pre-training document, and the Reasoning Score to measure the density and quality of reasoning chains present in the document. For the Education Score, a highly similar metric has already been adopted in QuRating [1], where the original LLM-as-a-judge prompt asks: “Which text has more educational value?” Our metric follows this established approach. We apologize for not citing this reference in Appendix J: LLM-AS-A-JUDGE EXPERIMENT DETAILS; we mistakenly assumed it to be a common practice. We will revise this in the camera-ready version.
> > >
> > > The Reasoning Score is a newly defined metric in our work, motivated by the fact that our method focuses on identifying documents with higher reasoning density. For example, we consider a math problem involving multiple lines of solution derivations to contain higher reasoning density than a purely textual biography of a mathematician.
> > > Since GPT-4o, which we use as the LLM judge, is a highly capable model with strong generalization ability and thus simple prompts are sufficient and do not require additional complexity or overly precise definitions, we intentionally kept the prompts simple rather than making them more complex or rigidly defined.
> > >
> > >
> > > 2. To verify the reliability of GPT-4o as an LLM judge for these metrics, we performed a human quality check before the submission of this paper. Among 20 sampled annotations, only 2 disagreed with human judgment, implying a 90% agreement rate between GPT-4o and human annotators, who were the two authors of this paper. Due to space limitations, we did not include these results in the paper; however, if necessary, we can provide an anonymized Google Spreadsheet containing these 20 human-evaluated samples to further demonstrate the reliability of GPT-4o as an LLM judge.
> > >
> > >
> > > 3. Regarding the reviewer’s observation that the Education Scores in Table 10 and Table 11 appear saturated, this phenomenon is primarily due to the high quality of the SmolLM corpus itself. This indirectly supports the reliability of the Education Score metric: when evaluating a corpus with uniformly high quality, the score naturally exhibits limited spread. Large-scale pre-training datasets of similar size but lower average quality are relatively scarce, which is why we ultimately chose the SmolLM corpus. The high quality of this corpus may also limit the headroom for improvement from data selection methods, which we believe partly explains why our gains over the baseline are around 1 percentage point. A larger and lower-quality corpus would likely amplify the benefits of our proposed method; however, this would incur substantially higher computational costs, and we leave such experiments for future work.
> > >
> > >
> > >
> > > [1] Wettig et al. (2024). QuRating: Selecting High-Quality Data for Training Language Models.

---

> ### Author Response · Authors · 2025-11-24
>
> Dear reviewer 3xfu,
>
> We appreciate the constructive feedback from you and the time you spent reviewing our work.
>
> We have carefully considered your comments and have endeavored to address these concerns in the response. As the discussion session draws to a close, we would greatly appreciate your guidance on whether our rebuttal adequately resolves your concerns. Additionally, we are more than willing to respond to any inquiries and address any feedback you may have.
>
> Thanks for your time and consideration.

---

### Author Response · Authors · 2025-12-01
**[1/2] Final Response to Area Chair regarding Paper AttentionInfluence: Adopting Attention Head Influence for Weak-to-Strong Pretraining Data Selection**

## Part A: Acknowledgment and Appreciation

We understand that due to the recent data leak and the subsequent reassignment of Area Chairs, you have stepped into this role under unusual and demanding circumstances. We sincerely appreciate the time and effort you are dedicating to reviewing our work and the prior discussions from scratch. We have summarized the reviewers' feedback and our rebuttals below to assist you in making an informed decision efficiently.

## Part B: Summary of Strengths

The reviewers generally responded positively to the novelty and efficiency of our approach. Key strengths highlighted across the reviews include:

* **Novelty and Perspective:** The paper introduces a new perspective by leveraging mechanistic interpretability (specifically retrieval head behavior) for pretraining data selection, moving beyond standard language modeling metrics (Reviewer 3xfu , Reviewer wvBs ).
* **Efficiency and "Weak-to-Strong" Generalization:** The method is praised for being training-free, unsupervised, and capable of using a small model (1.3B) to select data that improves a larger model (7B) (Reviewer 3xfu , Reviewer wvBs , Reviewer S3Rg ).
* **Metric Design:** The design of the AttentionInfluence Score was noted as convincing for data selection (Reviewer 7Ecp ).
* **Comprehensive Analysis:** Reviewers appreciated the detailed ablations, qualitative analyses, and the grounding in interpretability literature (Reviewer 3xfu , Reviewer wvBs ).
* **Performance:** The method outperforms relevant baselines on knowledge-intensive and reasoning-heavy benchmarks (Reviewer 3xfu , Reviewer wvBs , Reviewer 7Ecp ).

## Part C: Addressing Concerns

We have addressed the specific concerns raised by reviewers during the rebuttal phase. Below is a summary of the resolutions and where they can be found in the revised manuscript.

**1. Mismatch Between Retrieval Heads and Reasoning Tasks**
* **C1. The Concern:** Reviewer 3xfu  expressed concern that retrieval heads are typically associated with long-context tasks, while our evaluation focused on short-context reasoning tasks.
* **C2. Our Resolution:** We clarified that retrieval heads are mechanistically indispensable for Chain-of-Thought (CoT) reasoning. As established in prior work (Wu et al., 2024), masking these heads causes CoT performance to collapse. Therefore, these heads are a valid signal for selecting reasoning-intensive data. Furthermore, we provided internal results showing our method also improves long-context performance (NIAH extrapolation) , and we noted that the DROP benchmark used in our paper is effectively a long-context task (avg >1024 tokens).
* **C3. Location in Manuscript:** Appendix E (Mirror Effects in AttentionInfluence); Discussion on retrieval heads relation to CoT in Introduction and Section 4.1.

**2. Robustness and Applicability to Post-Training/Mid-Training**
* **C1. The Concern:** Reviewers wvBs  and S3Rg  asked if the method is robust across different models and if it applies to mid-training or post-training (SFT) stages.
* **C2. Our Resolution:**
    * **Post-Training:** We conducted additional experiments applying AttentionInfluence to the s1 dataset for SFT on Qwen2.5-32B-Instruct. Our selection outperformed the baseline s1K-1.1 on math benchmarks (AIME, MATH 500).
    * **Mid-Training:** Our main experiment simulates mid-training via a learning rate decay phase (746B-1000B tokens) where our method consistently outperforms baselines.
    * **Cross-Model Robustness:** We verified that using a 1.3B model to select data for a 7B model works (Weak-to-Strong) and internal tests confirmed dense models can select for MoE models.
* **C3. Location in Manuscript:** Appendix I (Detailed Performance Evolution); Appendix P (Limitations and Opportunities - Post-training discussion).

---

> ### Author Response · Authors · 2025-12-01
> **[2/2] Final Response to Area Chair regarding Paper AttentionInfluence: Adopting Attention Head Influence for Weak-to-Strong Pretraining Data Selection**
>
> **3. Magnitude of Gains and Baseline Comparisons**
> * **C1. The Concern:** Reviewer 3xfu  noted that gains were sometimes small (<1%) compared to the FineWeb-Edu classifier. Reviewers S3Rg  and 7Ecp  questioned the fairness of the data mixing strategy.
> * **C2. Our Resolution:**
>     * **Baselines:** FineWeb-Edu is a strong *supervised* baseline (distilled from Llama-70B). Our method is fully *unsupervised* and training-free, yet achieves comparable or better results at 1T tokens.
>     * **Fairness:** We ensured a strictly fair comparison. All baselines (including the main Baseline) utilize the same total token budget (1T), mixing the base corpus with the selected subset (or random samples for the baseline) in identical proportions.
> * **C3. Location in Manuscript:** Section 5.1 (Experimental Details); Table 2 (Final Results).
>
> **4. Definition of Analysis Metrics (Education & Reasoning Scores)**
> * **C1. The Concern:** Reviewer 3xfu  felt the "Education Score" and "Reasoning Score" used for analysis were loosely defined.
> * **C2. Our Resolution:** We clarified that the Education Score follows the established protocol from QuRating (Wettig et al., 2024). The Reasoning Score was validated via a human quality check, achieving a 90% agreement rate between the LLM judge and human annotators.
> * **C3. Location in Manuscript:** Section 6.1 (Reliability of AttentionInfluence); Appendix J (LLM-as-a-judge Experiment Details).
>
> **5. Technical Specifics (Head Detection & Learning Rate Decay)**
> * **C1. The Concern:** Reviewer 7Ecp  questioned the use of synthetic samples for head detection and the choice of the top 5% of heads. They also missed the Learning Rate Decay (LRD) results.
> * **C2. Our Resolution:**
>     * **Synthetic Data:** We utilize random hash strings in synthetic samples to ensure diversity and avoid data leakage, focusing purely on the retrieval mechanism.
>     * **Top 5%:** We follow the standard set by the original Retrieval Head paper (Wu et al., 2024), which identified the top 5% as critical.
>     * **LRD:** Table 1 shows results *without* LRD (mid-training), but Table 2 explicitly reports results *with* LRD after full training.
> * **C3. Location in Manuscript:** Section 4.1 (Method); Appendix A (Synthetic Test Sample); Table 2.

---

### Meta-Review · Area_Chair_oN9F · 2026-01-03

**Summary:**

This paper provides a data selection mechanism based on attention scores.

Reviewers have raised several important weaknesses:

W1) The main methodology relies on retrieval-head scores, which are more important for long-context tasks, whereas the downstream tasks evaluated are mostly short-context.

W2) The results are kind of weak compared to baselines.

W3) The Education score and Reasoning score are weakly defined metrics for evaluation.

W4) Only one pretraining corpus (SmolLM) and one pretraining model (a 7B model) are used.

W5) The samples used to identify the important attention heads are limited and do not have much discussion.

W6) Two reviewers have the concern that the training protocol samples a subset of the original pre-training dataset and combines it with the pre-training dataset. That seems to just unsample the data.

Additional questions: Studies on the mid- and post-training stages. Concern about search heads. Whether the screening model and training model need to be from the same series. Question on whether the method applies to continual pretraining

**Reviewer Concerns:**

The authors have provided a quite thorough rebuttal.

W1) The authors justified that the long-context tasks and reasoning capabilities are indeed related. But I don’t think the authors directly answer the short-context question. They also mentioned that their evaluations are not included in this submission and are mostly internal. I think the authors provided some quite understandable reasons for focusing on short-context questions only, but those also seem to confirm the reviewers’ concerns.

W2) While the authors demonstrated that their internal evaluations do show significant improvement on long-context tasks, that seems to consolidate the reviewer’s arguments.

W3) The authors provided sources for these scores.

W4) The authors provided results on an in-house SFT dataset.

W5) The authors provided an example, and they seem to have enough details on how these samples are generated.

W6) I don’t think the authors provide a great answer. The reviewer’s concerns do seem legitimate. It might be useful if the authors could provide a more thorough justification in the revised paper.

Additiona questionsl: The authors provided some experiments and convincing clarifications.

**Reviewer Scores:**

I would say that, if the reviewers had engaged more deeply, the scores could have improved from 4-6-2-4 to something like 4-6-4-6, as the authors addressed quite a few of the concerns. Unfortunately, that does not seem to have happened. I believe the authors can incorporate all of the feedback to significantly strengthen their next submission.

---

### Decision · Program_Chairs · 2026-01-26

Reject